# Average-Reward Learning and Planning with Options

**Yi Wan[†], Abhishek Naik[†], Richard S. Sutton[†‡]**
{wan6,anaik1,rsutton}@ualberta.ca
[†]University of Alberta, Amii      [‡]DeepMind
Edmonton, Canada      Edmonton, Canada

## Abstract

We extend the options framework for temporal abstraction in reinforcement learning from discounted Markov decision processes (MDPs) to average-reward MDPs. Our contributions include general convergent off-policy inter-option learning algorithms, intra-option algorithms for learning values and models, as well as sample-based planning variants of our learning algorithms. Our algorithms and convergence proofs extend those recently developed by Wan, Naik, and Sutton. We also extend the notion of option-interrupting behavior from the discounted to the average-reward formulation. We show the efficacy of the proposed algorithms with experiments on a continuing version of the Four-Room domain.

## 1 Introduction

Reinforcement learning (RL) is a formalism of trial-and-error learning in which an agent interacts with an environment to learn a behavioral strategy that maximizes a notion of reward. In many problems of interest, a learning agent may need to predict the consequences of its actions over multiple levels of temporal abstraction. The *options* framework provides a way for defining courses of actions over extended time scales, and for learning, planning, and representing knowledge with them (Sutton, Precup, & Singh 1999, Sutton & Barto 2018). The options framework was originally proposed within the *discounted* formulation of RL in which the agent tries to maximize the expected discounted return from each state. We extend the options framework from the discounted formulation to the *average-reward* formulation in which the goal is to find a policy that maximizes the rate of reward.

The average-reward formulation is of interest because, once genuine function approximation is introduced, there is no longer a well-defined discounted formulation of the continuing RL problem (see Sutton & Barto 2018, Section 10.4; Naik et al. 2019). If we want to take advantage of options in acting, learning, and planning in the continuing (non-episodic) RL setting, then we must extend options to the average-reward formulation.

Given a Markov decision process (MDP) and a fixed set of options, learning and planning algorithms can be divided into two classes. The first class consists of *inter*-option algorithms, which enable an agent to learn or plan with options instead of primitive actions. Given an option, the learning and planning updates for this option in these algorithms occur only *after* the option's actual or simulated execution. Algorithms in this class are also called semi-MDP (SMDP) algorithms because given an MDP, the decision process that selects among a set of options, executing each to termination, is an SMDP (Sutton et al. 1999). The second class consists of algorithms in which learning or planning updates occur after each state-action transition *within* options' execution — these are called *intra*-option algorithms. From a single state-action transition, these algorithms can learn or plan to improve the values or policies for *all* options that may generate that transition, and are therefore potentially more efficient than SMDP algorithms.

Several inter-option (SMDP) learning algorithms have been proposed for the average-reward formulation (see, e.g., Das et al. 1999, Gosavi 2004, Vien & Chung 2008). To the best of our knowledge,

35th Conference on Neural Information Processing Systems (NeurIPS 2021).

Gosavi's (2004) algorithm is the only proven-convergent *off-policy* inter-option learning algorithm. However, its convergence proof requires the underlying SMDP to have a special state that is recurrent under all stationary policies. Recently, Wan, Naik, and Sutton (2021) proposed Differential Q-learning, an off-policy control learning algorithm for average-reward MDPs that is proved to converge without requiring any special state. We extend this algorithm and its convergence proof from primitive actions to options and highlight some challenges we faced in developing *inter-option Differential Q-learning*. For planning, we propose *inter-option Differential Q-planning*, which is the first convergent *incremental* (sampled-based) planning algorithm. The existing proven-convergent inter-option planning algorithms (e.g., Schweitzer 1971, Puterman 1994, Li & Cao 2010) are not incremental because they perform a full sweep over states for each planning step.

Additionally, the literature lacks intra-option learning and planning algorithms within the average-reward formulation for both values and models. We fill this gap by proposing such algorithms in the average-reward formulation and provide their convergence results. These algorithms are stochastic approximation algorithms solving the average-reward intra-option value and model equations, which are also introduced in this paper for the first time.

Sutton et al. (1999) also introduced an algorithm to improve an agent's behavior given estimated option values. Instead of letting an option execute to termination, this algorithm involves potentially interrupting an option's execution to check if starting a new option might yield a better expected outcome. If so, then the currently-executing option is terminated, and the new option is executed. Our final contribution involves extending this notion of an *interruption* algorithm from the discounted to the average-reward formulation.

## 2   Problem Setting

We formalize an agent's interaction with its environment by a finite Markov decision process (MDP) $\mathcal{M}$ and a finite set of options $\mathcal{O}$. The MDP is defined by the tuple $\mathcal{M} \doteq (\mathcal{S}, \mathcal{A}, \mathcal{R}, p)$, where $\mathcal{S}$ is a set of states, $\mathcal{A}$ is a set of actions, $\mathcal{R}$ is a set of rewards, and $p : \mathcal{S} \times \mathcal{R} \times \mathcal{S} \times \mathcal{A} \to [0, 1]$ is the dynamics of the environment. Each option $o$ in $\mathcal{O}$ has two components: the option's *policy* $\pi^o : \mathcal{A} \times \mathcal{S} \to [0, 1]$, and a probability distribution of the option's *termination* $\beta^o : \mathcal{S} \to [0, 1]$. For simplicity, for any $s \in \mathcal{S}, o \in \mathcal{O}$, we use $\pi(a \mid s, o)$ to denote $\pi^o(a, s)$ and $\beta(s, o)$ to denote $\beta^o(s)$. Sutton et al.'s (1999) options additionally have an *initiation* set that consists of the states at which the option can be initiated. To simplify the presentation in this paper, we allow all options to be initiated in all states of the state space; the algorithms and theoretical results can easily be extended to incorporate initiation from specific states.

In the continuing (non-episodic) setting, the agent-environment interactions go on forever without any resets. If an option $o$ is initiated at time $t$ in state $S_t$, then the action $A_t$ is chosen according to the option's policy $\pi(\cdot \mid S_t, o)$. The agent then observes the next state $S_{t+1}$ and reward $R_{t+1}$ according to $p$. The option terminates at $S_{t+1}$ with probability $\beta(S_{t+1}, o)$ or continues with action $A_{t+1}$ chosen according to $\pi(\cdot \mid S_{t+1}, o)$. It then possibly terminates in $S_{t+2}$ according to $\beta(S_{t+2}, o)$, and so on. At an option termination, one way to govern an agent's behavior is to choose a new option according to a hierarchical policy $\mu_b : \mathcal{S} \times \mathcal{O} \mapsto [0, 1]$. In this case, when an option terminates at time $t$, the next option is selected stochastically according to $\mu_b(\cdot | S_t)$. The option initiates at $S_t$ and terminates at $S_{t+K}$, where $K$ is a random variable denoting the number of time steps the option executed. At $S_{t+K}$, a new option is again chosen according to $\mu_b(\cdot | S_{t+K})$, and so on. We use the notation $O_t$ to denote whatever option is being executed at time step $t$. Note that $O_t$ will remain the same for as many steps as the option executes. Also note that actions are a special case of options: every action $a$ is an option $o$ that terminates after exactly one step ($\beta(s, o) = 1, \ \forall s$) and whose policy is to pick $a$ in every state ($\pi(a \mid s, o) = 1, \ \forall s$).

Let $T_n$ denote the time step when the $n - 1^{\text{th}}$ option terminates and the $n^{\text{th}}$ option is chosen. Denote the $n^{\text{th}}$ option by $\hat{O}_n \doteq O_{T_n}$, its starting state by $\hat{S}_n \doteq S_{T_n}$, the cumulative reward during its execution by $\hat{R}_n \doteq \sum_{t=T_n+1}^{T_{n+1}} R_t$, the state it terminates in by $\hat{S}_{n+1} \doteq S_{T_{n+1}}$, and its length by $\hat{L}_n \doteq T_{n+1} - T_n$. Note that every option's length is a random variable taking values among positive integers. The option's transition probability is then defined as $\hat{p}(s', r, l \mid s, o) \doteq \Pr(\hat{S}_{n+1} = s', \hat{R}_n = r, \hat{L}_n = l \mid \hat{S}_n = s, \hat{O}_n = o)$. Throughout the paper, we assume that the expected execution time of every option starting from any state is finite.

An MDP $\mathcal{M}$ and a set of options $\mathcal{O}$ results in an SMDP $\hat{\mathcal{M}} = (\mathcal{S}, \mathcal{O}, \hat{\mathcal{L}}, \hat{\mathcal{R}}, \hat{p})$, where $\hat{\mathcal{L}}$ is the set of all possible lengths of options and $\hat{\mathcal{R}}$ is the set of all possible options' cumulative rewards. For this SMDP, the *reward rate* of a policy $\mu$ given a starting state $s$ and option $o$ can be defined as $r^C(\mu)(s, o) \doteq \lim_{t \to \infty} \mathbb{E}_\mu[\sum_{i=1}^t R_i \mid S_0 = s, O_0 = o]/t$. Alternatively, at the level of option transitions, $r(\mu)(s, o) \doteq \lim_{n \to \infty} \mathbb{E}_\mu[\sum_{i=0}^n \hat{R}_i \mid \hat{S}_0 = s, \hat{O}_0 = o]/\mathbb{E}_\mu[\sum_{i=0}^n \hat{L}_i \mid \hat{S}_0 = s, \hat{O}_0 = o]$. Both the limits exist and are equivalent (Puterman's (1994) propositions 11.4.1 and 11.4.7) under the following assumption:

**Assumption 1.** *The Markov chain induced by any stationary policy in the MDP* $(\mathcal{S}, \mathcal{O}, \hat{\mathcal{R}}, p')$ *is unichain, where* $p'(s', r \mid s, o) \doteq \sum_l \hat{p}(s', r, l \mid s, o) \ \forall \ s', r, s, o.$

**Note:** In a unichain MDP, there could be some states that only occur a finite number of times in a single stream of experience. In other words, these states are *transient* under all stationary policies. Thus, their values can not be correctly estimated by *any* learning algorithm. However, this inaccurate value estimation is not a problem because the decisions made in these transient states do not affect the reward rate. We refer to the non-transient states as *recurrent* states and denote their set by $\mathcal{S}' \subseteq \mathcal{S}$.

Under Assumption 1, the reward rate does not depend on the starting state-option pair and hence we can denote it by just $r(\mu)$. The optimal reward rate can then be defined as $r_* \doteq \sup_{\mu \in \Pi} r(\mu)$, where $\Pi$ denotes the set of all policies. The differential option-value function for a policy $\mu$ is defined for all $s \in \mathcal{S}, o \in \mathcal{O}$ as $q_\mu(s, o) \doteq \mathbb{E}_\mu[R_{t+1} - r(\mu) + R_{t+2} - r(\mu) + \cdots \mid S_t = s, O_t = o]$. The *evaluation* and *optimality* equations for SMDPs, as given by Puterman (1994), are:

$$q(s, o) = \sum_{s', r, l} \hat{p}(s', r, l \mid s, o)\big(r - \bar{r} \cdot l + \sum_{o'} \mu(o'|s')q(s', o')\big), \tag{1}$$

$$q(s, o) = \sum_{s', r, l} \hat{p}(s', r, l \mid s, o)\big(r - \bar{r} \cdot l + \max_{o'} q(s', o')\big), \tag{2}$$

where $q$ and $\bar{r}$ denote estimates of the option-value function and the reward rate respectively. If Assumption 1 holds, the SMDP Bellman equations have a unique solution for $\bar{r}$ — $r(\mu)$ for evaluation and $r_*$ for control — and a unique solution for $q$ only up to a constant (Schweitzer & Federgruen 1978). Given an MDP and a set of options, the goal of the *prediction* problem is, for a given policy $\mu$, to find the reward rate $r(\mu)$ and the differential value function (possibly with some constant offset). The goal of the *control* problem is to find a policy that achieves the optimal reward rate $r_*$.

## 3   Inter-Option Learning and Planning Algorithms

In this section, we present our inter-option learning and planning, prediction and control algorithms, which extend Wan et al.'s (2021) differential learning and planning algorithms for average-reward MDPs from actions to options. We begin with the control learning algorithm and then move on to the prediction and planning algorithms.

Consider Wan et al.'s (2021) control learning algorithm:

$$Q_{t+1}(S_t, A_t) \doteq Q_t(S_t, A_t) + \alpha_t \delta_t, \quad \bar{R}_{t+1} \doteq \bar{R}_t + \eta \alpha_t \delta_t,$$

where $Q$ is a vector of size $|\mathcal{S} \times \mathcal{A}|$ that approximates a solution of $q$ in the Bellman optimality equation for MDPs, $\bar{R}$ is a scalar estimate of the optimal reward rate, $\alpha_t$ is a step-size sequence, $\eta$ is a positive constant, and $\delta_t$ is the temporal-difference (TD) error: $\delta_t \doteq R_t - \bar{R}_t + \max_a Q_t(S_{t+1}, a) - Q_t(S_t, A_t)$. The most straightforward inter-option extension of Differential Q-learning is:

$$Q_{n+1}(\hat{S}_n, \hat{O}_n) \doteq Q_n(\hat{S}_n, \hat{O}_n) + \alpha_n \delta_n, \tag{3}$$

$$\bar{R}_{n+1} \doteq \bar{R}_n + \eta \alpha_n \delta_n, \tag{4}$$

where $Q$ is a vector of size $|\mathcal{S} \times \mathcal{O}|$ that approximates a solution of $q$ in (2), $\bar{R}$ is a scalar estimate of $r_*$, $\alpha_n$ is *a* step-size sequence, and $\delta_n$ is the TD error:

$$\delta_n \doteq \hat{R}_n - \hat{L}_n \bar{R}_n + \max_o Q_n(\hat{S}_{n+1}, o) - Q_n(\hat{S}_n, \hat{O}_n). \tag{5}$$

Such an algorithm is prone to instability because the *sampled* option length $\hat{L}_n$ can be quite large, and any error in the reward-rate estimate $\bar{R}_n$ gets multiplied with the potentially-large option length.

Using small step sizes might make the updates relatively stable, but at the cost of slowing down learning for options of shorter lengths. This could make the choice of step size quite critical, especially when the range of the options' lengths is large and unknown. Alternatively, inspired by Schweitzer (1971), we propose scaling the updates by the *estimated* length of the option being executed:

$$Q_{n+1}(\hat{S}_n, \hat{O}_n) \doteq Q_n(\hat{S}_n, \hat{O}_n) + \alpha_n \delta_n / L_n(\hat{S}_n, \hat{O}_n), \tag{6}$$

$$\bar{R}_{n+1} \doteq \bar{R}_n + \eta \alpha_n \delta_n / L_n(\hat{S}_n, \hat{O}_n), \tag{7}$$

where $\alpha_n$ is a step-size sequence, $L_n(\cdot, \cdot)$ comes from an additional vector of estimates $L : \mathcal{S} \times \mathcal{O} \to \mathbb{R}$ that approximates the expected lengths of state-option pairs, updated from experience by:

$$L_{n+1}(\hat{S}_n, \hat{O}_n) \doteq L_n(\hat{S}_n, \hat{O}_n) + \beta_n(\hat{L}_n - L_n(\hat{S}_n, \hat{O}_n)), \tag{8}$$

where $\beta_n$ is an another step-size sequence. The TD-error $\delta_n$ in (6) and (7) is

$$\delta_n \doteq \hat{R}_n - L_n(\hat{S}_n, \hat{O}_n)\bar{R}_n + \max_o Q_n(\hat{S}_{n+1}, o) - Q_n(\hat{S}_n, \hat{O}_n), \tag{9}$$

which is different from (5) with the estimated expected option length $L_n(\hat{S}_n, \hat{O}_n)$ being used instead of the sampled option length $\hat{L}_n$. (6–9) make up our *inter-option Differential Q-learning* algorithm.

Similarly, our prediction learning algorithm, called *inter-option Differential Q-evaluation*, also has update rules (6–8) with the TD error:

$$\delta_n \doteq \hat{R}_n - L_n(\hat{S}_n, \hat{O}_n)\bar{R}_n + \sum_o \mu(o \mid \hat{S}_{n+1})Q_n(\hat{S}_{n+1}, o) - Q_n(\hat{S}_n, \hat{O}_n). \tag{10}$$

**Theorem 1** (Convergence of inter-option algorithms; informal)**.** *If Assumption 1 holds, step sizes are decreased appropriately, all state-option pairs $(s, o)$ in $\mathcal{S}'$ and $\mathcal{O}$ are visited for an infinite number of times, and the relative visitation frequency between any two pairs is finite:*

1. *inter-option Differential Q-learning (6–9) converges almost surely, $\bar{R}_n$ to $r_*$ and $Q_n(s, o)$ to a solution of $q(s, o)$ in (2) for all $s \in \mathcal{S}', o \in \mathcal{O}$, and $r(\mu_n)$ to $r_*$, where $\mu_n$ is a greedy policy w.r.t. $Q_n$,*
2. *inter-option Differential Q-evaluation (6–8, 10) converges almost surely, $\bar{R}_n$ to $r(\mu)$ and $Q_n(s, o)$ to a solution of of $q(s, o)$ in (1) for all $s \in \mathcal{S}', o \in \mathcal{O}$.*

The convergence proofs for the inter-option (as well as the subsequent intra-option) algorithms are based on a result that generalizes Wan et al.'s (2021) and Abounadi et al.'s (2001) proof techniques from primitive actions to options. We present this result in Appendix A.1; the formal theorem statements and proofs in Appendix A.2.

**Remark**: It is important for the scaling factor in the algorithm to be the expected option length $L_n(\hat{S}_n, \hat{O}_n)$ and not the sampled option length $\hat{L}_n$. Scaling the updates by the expected option lengths ensures that fixed points of the updates are the solutions of (2). This is not guaranteed to be true when using the sampled option length. We discuss this in more detail in Appendix C.1.

The inter-option planning algorithms for prediction and control are similar to the learning algorithms except that they use simulated experience generated by a (given or learned) model instead of real experience. In addition, they only have two update rules, (6) and (7), not (8), because the model provides the expected length of a given option from a given state (see Section 5 for a complete specification of option models). The planning algorithms and their convergence results are presented in Appendix A.2.

**Empirical Evaluation.** We tested our inter-option Differential Q-learning with Gosavi's (2004) algorithm as a baseline in a variant of Sutton et al.'s (1999) Four-Room domain (shown in Figure 1). The agent starts in the yellow cell. The goal states are indicated by green cells. Every experiment in this paper uses only one of the green cells as a goal state; the other two are considered as empty cells.

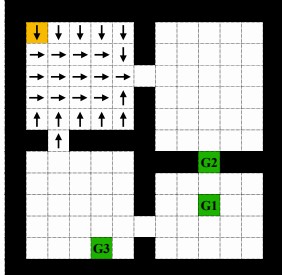

Figure 1: A continuing variant of the Four-Room domain where the task is to repeatedly go from the yellow start state to one of the three green goal states. There is one goal state per experiment, chosen to demonstrate particular aspects of the proposed algorithms. Also shown is an option policy to go to the upper hallway cell; more details in-text.

There are four primitive actions of moving up, down, left, right. The agent receives a reward of +1 when it moves into the goal cell, 0 otherwise.

In addition to the four primitive actions, the agent has eight options that take it from a given room to the hallways adjoining the room. The arrows in Figure 1 illustrate the policy of one of the eight options. For this option, the policy in the empty cells (not marked with arrows) is to uniformly-randomly pick among the four primitive actions. The termination probability is 0 for all the cells with arrows and 1 for the empty cells. The other seven options are defined in a similar way. Denote the set of primitive actions as $\mathcal{A}$ and the set of hallway options as $\mathcal{H}$. Including the primitive actions, the agent has 12 options in total.

In the first experiment, we tested inter-option Differential Q-learning with three different sets of options, $\mathcal{O} \in \{\mathcal{A}, \mathcal{H}, \mathcal{A} + \mathcal{H}\}$. The task was to reach the green cell G1, which the agent can achieve with a combination of options and primitive actions. The shortest path to G1 from the starting state takes 16 time steps, hence the best possible reward rate for this task is $1/16 = 0.0625$. The agent used an $\epsilon$-greedy policy with $\epsilon = 0.1$. For each of the two step-sizes $\alpha_n$ and $\beta_n$, we tested five choices: $2^{-x}, x \in \{1, 3, 5, 7, 9\}$. In addition, we tested five choices of $\eta : 10^{-x}, x \in \{0, 1, 2, 3, 4\}$. $Q$ and $\bar{R}$ were initialized to 0, $L$ to 1. Each parameter setting was run for 200,000 steps and repeated 30 times. The left subfigure of Figure 2 shows a typical learning curve for each of the three sets of options, with $\alpha = 2^{-3}$, $\beta = 2^{-1}$, and $\eta = 10^{-1}$. The parameter study for $\mathcal{O} = \mathcal{A} + \mathcal{H}$ w.r.t. $\alpha$ and $\eta$, with $\beta = 2^{-1}$, is presented in the right subfigure of Figure 2. The metric is the average reward obtained over the entire training period. Complete parameter studies for all the three sets of options are presented in Appendix B.1.

The learning curves in the left panel of Figure 2 show that the agent achieved a relatively stable reward rate after 100,000 steps in all three cases. Using just primitive actions $\mathcal{A}$, the learning curve rises the slowest, indicating that hallway options indeed help the agent reach the goal faster. But solely using the hallway options $\mathcal{H}$ is not very useful in the long run as the goal G1 is not a hallway state. Note that because of the $\epsilon$-greedy behavior policy, the learning curves do not reach the optimal reward rate of 0.0625. These observations mirror those by Sutton et al. (1999) in the discounted formulation.

The sensitivity curves of inter-option Differential Q-learning (right panel of Figure 2) indicate that, in this Four-Room domain, the algorithm was not sensitive to parameter $\eta$, performed well for a wide range of step sizes $\alpha$, and showed low variance across different runs. We also found that the algorithm was not sensitive to $\beta$ either; this parameter study is also presented in Appendix B.1.

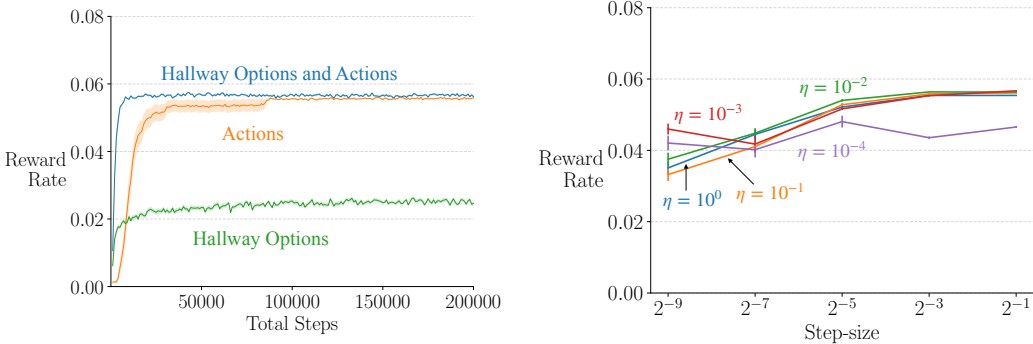

Figure 2: Plots showing some learning curves and the parameter study of inter-option Differential Q-learning on the continuing Four-Room domain when the goal was to go to G1. *Left*: A point on the solid line denotes reward rate over the last 1000 time steps and the shaded region indicates one standard error. The behavior using the three different sets of options was as expected. *Right*: Sensitivity of performance to $\alpha$ and $\eta$ when using $\mathcal{O} = \mathcal{A} + \mathcal{H}$ and $\beta = 2^{-1}$. The x-axis denotes step size $\alpha$; the y-axis denotes the rate of the rewards averaged over all 200,000 steps of training, reflecting the rate of learning. The error bars denote one standard error. The algorithm's rate of learning varied little over a broad range of $\eta$.

We also tested Gosavi's (2004) algorithm as a baseline. We chose not to compare the proposed algorithms in this paper with Sutton et al.'s (1999) discounted versions because the discounted and average-reward problem formulations are different; comparing the performance of their respective solution methods would be inappropriate and difficult to interpret. We have proposed new solution methods for the average-reward formulation, hence in this case Gosavi's (2004) algorithm is the most appropriate baseline. Recall it is the only proven-convergent average-reward SMDP off-policy control learning algorithm prior to our work. It estimates the reward rate by tracking the cumulative reward $\bar{C}$ obtained by the options and dividing it by the another estimate $\bar{T}$ the tracks the length of the options. If the $n^{\text{th}}$ option executed is a greedy choice, then these estimates are updated using:

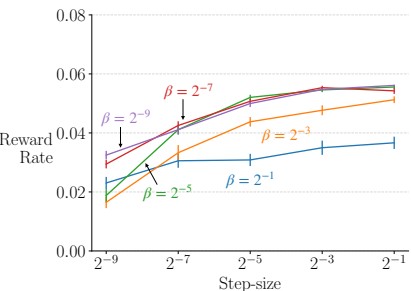

Figure 3: Parameter studies showing the baseline algorithm's (Gosavi 2004) rate of learning is relatively more sensitive to the choices of its two parameters compared to our inter-option Differential Q-learning. The experimental setting and the plot axes are the same as mentioned in Figure 2's caption.

$$\bar{C}_{n+1} \doteq \bar{C}_n + \beta_n(\hat{R}_n - \bar{C}_n),$$
$$\bar{T}_{n+1} \doteq \bar{T}_n + \beta_n(\hat{L}_n - \bar{T}_n),$$
$$\bar{R}_{n+1} \doteq \bar{C}_{n+1}/\bar{T}_{n+1}.$$

When $\hat{O}_n$ is not greedy, $\bar{R}_{n+1} = \bar{R}_n$. The option-value function is updated with (3) with $\delta_n$ as defined in (5). $\alpha_n$ and $\beta_n$ are two step-size sequences. The sensitivity of this algorithm with $\mathcal{O} = \mathcal{A} + \mathcal{H}$ is shown in Figure 3. Compared to inter-option Differential Q-learning, this baseline has one less parameter, but its performance was found to be more sensitive to the values of both its step-size parameters. In addition, the error bars were generally larger, suggesting that the variance across different runs was also higher.

To conclude, our experiments with the continuing Four-Room domain show that our inter-option Differential Q-learning indeed finds the optimal policy given a set of options, in accordance with Theorem 1. In addition, its performance seems more robust to the choices of parameters compared to the baseline. Experiments with the prediction algorithm, inter-option Differential Q-evaluation, are presented in Appendix B.4.

## 4   Intra-Option Value Learning and Planning Algorithms

In this section, we introduce intra-option value learning and planning algorithms. The objectives are same as that of inter-option value learning algorithms. As mentioned earlier, intra-option algorithms learn from every transition $S_t, A_t, R_{t+1}, S_{t+1}$ during the execution of a given option $O_t$. Moreover, intra-option algorithms also make updates for *every* option $o \in \mathcal{O}$, including ones that may potentially never be executed.

To develop our algorithms, we first establish the intra-option evaluation and optimality equations in the average-reward case. The general form of the intra-option Bellman equation is:

$$q(s, o) = \sum_a \pi(a \mid s, o) \sum_{s', r} p(s', r \mid s, a)\Big(r - \bar{r} + u^q(s', o)\Big) \tag{11}$$

where $q \in \mathbb{R}^{|\mathcal{S}| \times |\mathcal{O}|}$ and $\bar{r} \in \mathbb{R}$ are free variables. The optimality and evaluation equations use $u^q = u^q_*$ and $u^q = u^q_\mu$ respectively, defined $\forall\, s' \in \mathcal{S}, o \in \mathcal{O}$ as:

$$u^q(s', o) = u^q_*(s', o) \doteq \big(1 - \beta(s', o)\big)q(s', o) + \beta(s', o)\max_{o'} q(s', o'), \tag{12}$$

$$u^q(s', o) = u^q_\mu(s', o) \doteq \big(1 - \beta(s', o)\big)q(s', o) + \beta(s', o)\sum_{o'} \mu(o'|s')q(s', o'). \tag{13}$$

Intuitively, the $u^q$ term accounts for the two possibilities of an option terminating or continuing in the next state. These equations generalize the average-reward Bellman equations given by Puterman (1994). The following theorem characterizes the solutions to the intra-option Bellman equations.

**Theorem 2** (Solutions to intra-option Bellman equations). *If Assumption 1 holds, then:*

1. *a) there exists a $\bar{r} \in \mathbb{R}$ and a $q \in \mathbb{R}^{|\mathcal{S}| \times |\mathcal{O}|}$ for which (11) and (12) hold,*
   *b) the solution of $\bar{r}$ is unique and is equal to $r_*$, let $q_1$ be one solution of $q$, the solutions of $q$*
   *form a set $\{q_1 + ce \mid c \in \mathbb{R}\}$ where $e$ is an all-one vector of size $|\mathcal{S}| \times |\mathcal{O}|$,*
   *c) a greedy policy w.r.t. a solution of $q$ achieves the optimal reward rate $r_*$.*

2. *a) there exists a $\bar{r} \in \mathbb{R}$ and a $q \in \mathbb{R}^{|\mathcal{S}| \times |\mathcal{O}|}$ for which (11) and (13) hold,*
   *b) the solution of $\bar{r}$ is unique and is equal to $r(\mu)$, the solutions of $q$ form a set $\{q_\mu + ce \mid c \in \mathbb{R}\}$.*

The proof extends those of Corollary 8.2.7, Theorem 8.4.3, Theorem 8.4.4 by Puterman (1994) and is presented in Appendix A.3.

Our intra-option control and prediction algorithms are stochastic approximation algorithms solving the intra-option optimality and evaluation equations respectively. Both the algorithms maintain a vector of estimates $Q(s, o)$ and a scalar estimate $\bar{R}$, just like our inter-option algorithms. However, unlike inter-option algorithms, intra-option algorithms need not maintain an estimator for option lengths ($L$) because they make updates after every transition. Our control algorithm, called *intra-option Differential Q-learning*, updates the estimates $Q$ and $\bar{R}$ by:

$$Q_{t+1}(S_t, o) \doteq Q_t(S_t, o) + \alpha_t \rho_t(o) \delta_t(o), \quad \forall\, o \in \mathcal{O}, \tag{14}$$

$$\bar{R}_{t+1} \doteq \bar{R}_t + \eta \alpha_t \sum_{o \in \mathcal{O}} \rho_t(o) \delta_t(o), \tag{15}$$

where $\alpha_t$ is a step-size sequence, $\rho_t(o) \doteq \frac{\pi(A_t|S_t,o)}{\pi(A_t|S_t,O_t)}$ is the importance sampling ratio, and:

$$\delta_t(o) \doteq R_{t+1} - \bar{R}_t + u_*^{Q_t}(S_{t+1}, o) - Q_t(S_t, o). \tag{16}$$

Our prediction algorithm, called *intra-option Differential Q-evaluation*, also updates $Q$ and $\bar{R}$ by (14) and (15) but with the TD error:

$$\delta_t(o) \doteq R_{t+1} - \bar{R}_t + u_\mu^{Q_t}(S_{t+1}, o) - Q_t(S_t, o). \tag{17}$$

**Theorem 3** (Convergence of intra-option algorithms; informal). *Under the conditions of Theorem 1:*

1. *intra-option Differential Q-learning algorithm (14–16) converges almost surely, $\bar{R}_t$ to $r_*$, $Q_t(s, o)$ to a solution of $q(s, o)$ in (11) and (12) for all $s \in \mathcal{S}', o \in \mathcal{O}$, and $r(\mu_t)$ to $r_*$, where $\mu_t$ is a greedy policy w.r.t. $Q_t$,*

2. *intra-option Differential Q-evaluation algorithm (14,15,17) converges almost surely, $\bar{R}_t$ to $r(\mu)$, $Q_t(s, o)$ to a solution of $q(s, o)$ in (11) and (13) for all $s \in \mathcal{S}', o \in \mathcal{O}$.*

**Remark:** The intra-option learning methods introduced in this section can be used with options having stochastic policies. This is possible with the use of the important sampling ratios as described above. Sutton et al.'s (1999) discounted intra-option learning methods were restricted to options having deterministic policies.

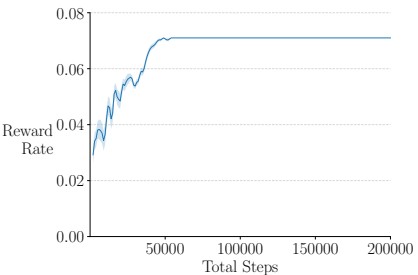

Again, the intra-option value planning algorithms are similar to the learning algorithms except that they use simulated experience generated by a given or learned model instead of real experience. The planning algorithms and their convergence results are presented in Appendix A.4.

**Empirical Evaluation**. We conducted another experiment in the Four-Room domain to show that intra-option Differential Q-learning can learn the values of hallway options $\mathcal{H}$ using only primitive actions $\mathcal{A}$. As mentioned earlier, there are no baseline intra-option average-reward algorithms, so this is a proof-of-concept experiment.

The goal state for this experiment was G2, which can be reached using the option that leads to the lower hallway.

Figure 4: Learning curve showing that the greedy policy corresponding to the hallway options' option-value function achieves the optimal reward rate on the continuing Four-Room domain. The value function was learned via intra-option Differential Q-learning using a behavior policy consisting only of primitive actions; the hallway options were never executed.

The optimal reward rate in this case is $1/14 \approx 0.714$ with both $\mathcal{O} = \mathcal{H}$ and $\mathcal{O} = \mathcal{A}$. We applied intra-option Differential Q-learning using a behavior policy that chose the four primitive actions with equal probability in all states. This choice of behavior policy and goal G2 would test if the intra-option algorithm leads to a policy consisting exclusively of options by interacting with the environment using only primitive actions. Each parameter setting was run for 200,000 steps and repeated 30 times. For evaluation, we saved the learned option value function after every 1000 steps and computed the average reward of the corresponding greedy policy over 1000 steps.

Figure 4 shows the learning curve of this average reward across the 30 independent runs for parameters $\alpha = 0.125, \eta = 0.1$. The agent indeed succeeds in learning the option-value function corresponding to the hallway options using a behavior policy consisting only of primitive actions. The parameter study of intra-option Differential Q-learning is presented in Appendix B.2. Experiments with the prediction algorithm, intra-option Differential Q-evaluation, are presented in Appendix B.4.

## 5   Intra-Option Model Learning and Planning Algorithms

In this section, we first describe option models within the average-reward formulation. We then introduce an algorithm to learn such models in an intra-option fashion. This option-model learning algorithm can be combined with the planning algorithms from the previous section to obtain a complete model-based average-reward options algorithm that learns option models and plans with them (we present this combined algorithm in Appendix C.2).

The average-reward option model is similar to the discounted options model but with key distinctions. Sutton et al.'s (1999) discounted option model has two parts: the dynamics part and the reward part. Given a state and an option, the dynamics part predicts the discounted occupancy of states upon termination, and the reward part predicts the expected (discounted) sum of rewards during the execution of the option. In the average-reward setting, apart from the dynamics and the reward parts, an option model has a third part — the *duration* part — that predicts the duration of execution of the option. In addition, the dynamics part predicts the state distribution upon termination without discounting and reward part predicts the undiscounted cumulative rewards during the execution of the option.

Formally, the dynamics part approximates $m^p(s'|s, o) \doteq \sum_{r,l} \hat{p}(s', r, l \,|s, o)$, the probability that option $o$ terminates in state $s'$ when starting from state $s$. The reward part approximates $m^r(s, o) \doteq \sum_{s',r,l} \hat{p}(s', r, l \,|s, o)\, r$, the expected cumulative reward during the execution of option $o$ when starting from state $s$. Finally, the duration part approximates $m^l(s, o) \doteq \sum_{s',r,l} \hat{p}(s', r, l \,|s, o)\, l$, the expected duration of option $o$ when starting from state $s$.

We now present a set of recursive equations that are key to our model-learning algorithms. These equations extend the discounted Bellman equations for option models (Sutton et al. 1999) to the average-reward formulation.

$$\bar{m}^p(x \mid s, o) = \sum_a \pi(a \mid s, o) \sum_{s',r} p(s', r \mid s, a)\Big(\beta(s', o)\mathbb{I}(x = s') + \big(1 - \beta(s', o)\big)\bar{m}^p(x \mid s', o)\Big), \quad (18)$$

$$\bar{m}^r(s, o) = \sum_a \pi(a \mid s, o) \sum_{s',r} p(s', r \mid s, a)\big(r + (1 - \beta(s', o))\bar{m}^r(s', o)\big), \quad (19)$$

$$\bar{m}^l(s, o) = \sum_a \pi(a \mid s, o) \sum_{s',r} p(s', r \mid s, a)\big(1 + (1 - \beta(s', o))\bar{m}^l(s', o)\big). \quad (20)$$

The first equation are different from the other two because the total reward and length of the option $o$ are incremented irrespective of whether the option terminates in $s'$ or not. The following theorem shows that $(m^p, m^r, m^l)$ is the unique solution of (18–20) and therefore the models can be obtained by solving these equations (see Appendix A.5 for the proof).

**Theorem 4** (Solutions to Bellman equations for option models). *There exist unique* $\bar{m}^p \in \mathbb{R}^{|\mathcal{S}| \times |\mathcal{O}| \times |\mathcal{S}|}$, $\bar{m}^r \in \mathbb{R}^{|\mathcal{S}| \times |\mathcal{O}|}$, *and* $\bar{m}^l \in \mathbb{R}^{|\mathcal{S}| \times |\mathcal{O}|}$ *for which* (18), (19), *and* (20) *hold. Further, if* $\bar{m}^p, \bar{m}^r, \bar{m}^l$ *satisfy* (18), (19), *and* (20), *then* $\bar{m}^p = m^p, \bar{m}^r = m^r, \bar{m}^l = m^l$.

Our *intra-option model-learning* algorithm solves the above recursive equations using the following TD-like update rules for each option $o$:

$$M_{t+1}^p(x \mid S_t, o) \doteq M_t^p(x \mid S_t, o) + \alpha_t \rho_t(o)\Big(\beta(S_{t+1}, o)\mathbb{I}(S_{t+1} = x)$$

$$+ \big(1 - \beta(S_{t+1}, o)\big)M_t^p(x \mid S_{t+1}, o) - M_t^p(x \mid S_t, o)\Big), \quad \forall\, x \in \mathcal{S}, \tag{21}$$

$$M_{t+1}^r(S_t, o) \doteq M_t^r(S_t, o) + \alpha_t \rho_t(o)\Big(R_{t+1} + \big(1 - \beta(S_{t+1}, o)\big)M_t^r(S_{t+1}, o) - M_t^r(S_t, o)\Big) \tag{22}$$

$$M_{t+1}^l(S_t, o) \doteq M_t^l(S_t, o) + \alpha_t \rho_t(o)\Big(1 + \big(1 - \beta(S_{t+1}, o)\big)M_t^l(S_{t+1}, o) - M_t^l(S_t, o)\Big) \tag{23}$$

where $M^p$ is a $|\mathcal{S}| \times |\mathcal{O}| \times |\mathcal{S}|$-sized vector of estimates, $M^r$ and $M^l$ are both $|\mathcal{S}| \times |\mathcal{O}|$-sized vectors of estimates, and $\alpha_t$ is a sequence of step sizes. Standard stochastic approximation results can be applied to show the algorithm's convergence (see Appendix A.6 for details).

**Theorem 5** (Convergence of the intra-option model learning algorithm; informal)**.** *If the step sizes are set appropriately and all the state-option pairs are updated an infinite number of times, then intra-option model-learning (21–23) converges almost surely, $M_t^p$ to $m^p$, $M_t^r$ to $m^r$, and $M_t^l$ to $m^l$.*

Our intra-option model-learning algorithms (21–23) can be applied with simulated one-step transitions generated by a *given action model*, resulting in a planning algorithm that produces an *estimated option model*. The planning algorithm and its convergence result are presented in Appendix A.6.

## 6 Interruption to Improve Policy Over Options

In all the algorithms we considered so far, the policy over options would select an option, execute the option policy till termination, then select a new option. Sutton et al. (1999) showed that the policy over options can be improved by allowing the *interruption* of an option midway through its execution to start a seemingly better option. We now show that this interruption result applies for average-reward options as well (see Appendix A.7 for the proof).

**Theorem 6** (Interruption)**.** *For any MDP, any set of options $\mathcal{O}$, and any policy $\mu : \mathcal{S} \times \mathcal{O} \to [0, 1]$, define a new set of options, $\mathcal{O}'$, with a one-to-one mapping between the two option sets as follows: for every $o = (\pi, \beta) \in \mathcal{O}$, define a corresponding $o' = (\pi, \beta') \in \mathcal{O}'$ where $\beta' = \beta$, but for any state $s$ in which $q_\mu(s, o) < v_\mu(s)$, $\beta'(s, o) = 1$ (where $v_\mu(s) \doteq \sum_o \mu(o \mid s)q_\mu(s, o)$). Let the interrupted policy $\mu'$ be such that for all $s \in \mathcal{S}$ and for all $o' \in \mathcal{O}'$, $\mu'(s, o') = \mu(s, o)$, where $o$ is the option in $\mathcal{O}$ corresponding to $o'$. Then:*

1. *the new policy over options $\mu'$ is not worse than the old one $\mu$, i.e., $r(\mu') \geq r(\mu)$,*

2. *if there exists a state $s \in \mathcal{S}$ from which there is a non-zero probability of encountering an interruption upon initiating $\mu'$ in $s$, then $r(\mu') > r(\mu)$.*

In short, the above theorem shows that interruption produces a behavior that achieves a higher reward rate than without interruption. Note that interruption behavior is only applicable with intra-option algorithms; complete option transitions are needed in inter-option algorithms.

**Empirical Evaluation.** We tested the intra-option Differential Q-learning algorithm with and without interruption in the Four-Room domain. We set the goal as `G3` and allowed the agent to choose and learn only from the set of all hallway options $\mathcal{H}$. With just hallway options, without interruption, the best strategy is to first move to the lower hallway and then try to reach the goal by following options that pick random actions in the states near the hallway and goal. With interruption, the agent can first move to the left hallway, then take the option that moves the agent to the lower hallway but terminate when other options have

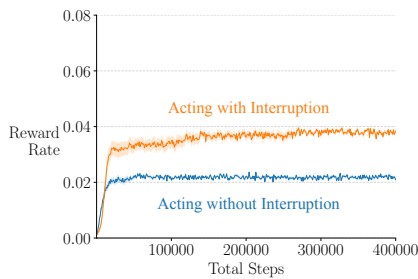

Figure 5: Learning curves showing that executing options with interruptions can achieve a higher reward rate than executing options till termination in the domain described in the adjoining text.

higher option-values. This termination is most likely to occur in the cell just above `G3`. The agent then needs a fewer number of steps in expectation to reach the goal.

Figure 5 shows learning curves using intra-option Differential Q-learning with and without interruptions on this problem. Each parameter setting was run for 400,000 steps and repeated 30 times. The learning curves shown correspond to $\alpha = 0.125$ and $\eta = 0.1$. As expected, the agent achieved a higher reward rate by using interruptions. The parameter study of the interruption algorithm along with the rest of the experimental details is presented in Appendix B.3.

# 7    Conclusions, Limitations, and Future Work

In this paper, we extended learning and planning algorithms for the options framework — originally proposed by Sutton et al. (1999) for discounted-reward MDPs — to average-reward MDPs. The inter-option learning algorithm presented in this paper is more general than previous work in that its convergence proof does not require existence of any special states in the MDP. We also derived the intra-option Bellman equations in average-reward MDPs and used them to propose the first intra-option learning algorithms for average-reward MDPs. Finally, we extended the interruption algorithm and its related theory from the discounted to the average-reward setting. Our experiments on a continuing version of the classic Four-Room domain demonstrate the efficacy of the proposed algorithms. We believe that our contributions will enable widespread use of options in the average-reward setting.

We now briefly comment on the novelty of our theoretical and algorithmic contributions. Our primary theoretical contribution is to generalize Wan et al.'s (2021) proof techniques to obtain a unified convergence proof for actions and options. The same proof techniques then apply for both the inter- and intra-option algorithms. Our primary algorithmic contribution is the scaling of the updates by option lengths in the inter-option algorithms. The lack of scaling makes the algorithms unstable and prone to divergence. Furthermore, we show the correct way of scaling involves estimated option lengths, not sampled option lengths.

The most immediate line of future work involves extending these ideas from the tabular case to the general case of function approximation, starting with linear function approximation. One way to incorporate function approximation is to extend algorithms presented in this paper to those using linear options (Sorg & Singh 2010, Yao et al. 2014), perhaps by building on Zhang et al.'s (2021) work. Using the results developed in this paper, we also foresee extensions to more ideas from the discounted formulation involving function approximation, such as Bacon et al.'s (2017) option-critic architecture, to the average-reward formulation.

This paper assumes that a fixed set of options is provided and the agent then learns or plans using them. One of the most important challenges in the options framework is the *discovery* of options. We think the discovery problem is orthogonal to the problem formulation. Hence, another line of future work is to extend existing option-discovery algorithms developed for the discounted formulation to the average-reward formulation (e.g., algorithms by McGovern & Barto 2001, Menache et al. 2002, Şimşek & Barto 2004, Singh et al. 2004, Van Djik & Polani 2011, Machado et al. 2017) . Relatively more work might be required in extending approaches that couple the problems of option discovery and learning (e.g., Gregor et al. 2016, Eysenbach et al. 2018, Achiam et al. 2018, Veeriah et al. 2021).

Another limitation of this paper is that it deals with learning and planning separately. We also need combined methods that learn models as well as plan with them; we discuss some ideas in Appendix C. Finally, we would like to get more empirical experience with the algorithms proposed in this paper, both in pedagogical tabular problems and challenging large-scale problems. Nevertheless, we believe this paper makes novel contributions that are significant for the use of temporal abstractions in average-reward reinforcement learning.

# Acknowledgements

The authors were supported by DeepMind, Amii, and CIFAR. The authors wish to thank Huizhen Yu for extensive discussions on several related works; Benjamin Van Roy, Csaba Szepesvári, Dale Schuurmans, Martha White, and the anonymous reviewers for valuable feedback.

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
