# A Formal Theoretical Results and Proofs

In this section, we provide formal statements of the theorems presented in the main text of the paper and show their proofs. This section has several subsections. The first subsection introduces General RVI Q, which will be used in later subsections. The other six subsections correspond to six theorems presented in the main text.

## A.1 General RVI Q

Wan et al. (2021) extended the family of RVI Q-learning algorithms (Abounadi, Bertsekas, and Borkar et al. 2001) to prove the convergence of their Differential Q-learning algorithm. Unlike RVI Q-learning, Differential Q-learning does not require a reference function. We further extend Wan et al.'s extended family of RVI Q-learning algorithms to a more general family of algorithms, called *General RVI Q*. We then prove convergence for this family of algorithms and show that inter-option algorithms and intra-option value learning algorithms are all members of this family.

We first need the following definitions:

1. a set-valued process $\{Y_n\}$ taking values in the set of nonempty subsets of $\mathcal{I}$ with the interpretation: $Y_n = \{i : i^{\text{th}}$ component of $Q$ was updated at time $n\}$,
2. $\nu(n, i) \doteq \sum_{k=0}^{n} I\{i \in Y_k\}$, where $I$ is the indicator function. Thus $\nu(n, i) =$ the number of times the $i$ component was updated up to step $n$,
3. i.i.d. random vectors $R_n$, $G_n$ and $F_n$ for all $n \geq 0$ satisfying $\mathbb{E}[R_n(i)] = r(i)$, where $r$ is a fixed real vector, $\mathbb{E}[G_n(Q)(i)] = g(Q)(i)$ for any $Q \in \mathbb{R}^{|\mathcal{I}|}$ where $g : \mathcal{I} \to \mathcal{I}$ is a function satisfying Assumption A.1 and $\mathbb{E}[F_n(Q)(i)] = f(Q)$ for any $i \in \mathcal{I}$ and $Q \in \mathbb{R}^{|\mathcal{I}|}$ where $f : \mathcal{I} \to \mathbb{R}$ is a function satisfying Assumption A.2.

**Assumption A.1.** *1) $g$ is a max-norm non-expansion, 2) $g$ is a span-norm non-expansion, 3) $g(x + ce) = g(x) + ce$ for any $c \in \mathbb{R}, x \in \mathbb{R}^{|\mathcal{I}|}$, 4) $g(cx) = cg(x)$ for any $c \in \mathbb{R}, x \in \mathbb{R}^{|\mathcal{I}|}$.*

**Assumption A.2.** *1) $f$ is L-Lipschitz, 2) there exists a positive scalar $u$ s.t. $f(e) = u$ and $f(x+ce) = f(x) + cu$, 3) $f(cx) = cf(x)$.*

**Assumption A.3.** *For $n \in \{0, 1, 2, \dots\}$, $\mathbb{E}[\|R_n - r\|^2] \leq K$, $\mathbb{E}[\|G_n(Q) - g(Q)\|^2] \leq K(1 + \|Q\|^2)$ for any $Q \in \mathbb{R}^{|\mathcal{I}|}$, and $\mathbb{E}[\|F_n(Q) - f(Q)e\|^2] \leq K(1 + \|Q\|^2)$ for any $Q \in \mathbb{R}^{|\mathcal{I}|}$ for a suitable constant $K > 0$.*

The above assumption means that the variances of $R_n$, $G_n(Q)$, and $F_n(Q)$ for any $Q$ are bounded.

General RVI Q's update rule is

$$Q_{n+1}(i) \doteq Q_n(i) + \alpha_{\nu(n,i)}\big(R_n(i) - F_n(Q_n)(i) + G_n(Q_n)(i) - Q_n(i) + \epsilon_n(i)\big)I\{i \in Y_n\},$$
(A.1)

where $\alpha_{\nu(n,i)}$ is the stepsize and $\epsilon_n$ is a sequence of random vectors of size $|\mathcal{I}|$.

We make following assumption on $\epsilon_n$.

**Assumption A.4** (Noise Assumption). *$\|\epsilon_n\|_\infty \leq K(1 + \|Q_n\|_\infty)$ for some scalar $K$. Further, $\epsilon_n$ converges in probability to 0.*

We make following assumptions on $\alpha_{\nu(n,i)}$.

**Assumption A.5** (Stepsize Assumption). *For all $n \geq 0$, $\alpha_n > 0$, $\sum_{n=0}^{\infty} \alpha_n = \infty$, and $\sum_{n=0}^{\infty} \alpha_n^2 < \infty$.*

**Assumption A.6** (Asynchronous Stepsize Assumption A). *Let $[\cdot]$ denote the integer part of $(\cdot)$, for $x \in (0, 1)$,*

$$\sup_i \frac{\alpha_{[xi]}}{\alpha_i} < \infty$$

*and*

$$\frac{\sum_{j=0}^{[yi]} \alpha_j}{\sum_{j=0}^{i} \alpha_j} \to 1$$

*uniformly in $y \in [x, 1]$.*

**Assumption A.7** (Asynchronous Stepsize Assumption B). *There exists $\Delta > 0$ such that*

$$\liminf_{n\to\infty} \frac{\nu(n,i)}{n+1} \geq \Delta,$$

*a.s., for all $s \in \mathcal{S}, o \in \mathcal{O}$. Furthermore, for all $x > 0$, let*

$$N(n,x) = \min\left\{ m > n : \sum_{i=n+1}^{m} \alpha_i \geq x \right\},$$

*the limit*

$$\lim_{n\to\infty} \frac{\sum_{i=\nu(n,i)}^{\nu(N(n,x),i)} \alpha_i}{\sum_{i=\nu(n,i')}^{\nu(N(n,x),i')} \alpha_i}$$

*exists a.s. for all $s, s', o, o'$.*

**Assumption A.8.** $r(i) - \bar{r} + g(q)(i) - q(i) = 0, \forall i \in \mathcal{I}$ *has a unique solution for $\bar{r}$ and a unique for $q$ only up to a constant.*

Denoted the unique solution of $\bar{r}$ by $r_\infty$. Further, it can be seen that the solution of $q$ satisfying both $r - \bar{r}e - g(q) - q = 0$ and $f(q) = r_\infty$ is unique because our assumption on $f$ (Assumption A.2). Denote the unique solution as $q_\infty$. We have,

$$f(q_\infty) = r_\infty. \tag{A.2}$$

**Theorem A.1.** *Under Assumptions A.1-A.8, General RVI Q converges, almost surely, $Q_n$ to $q_\infty$ and $f(Q_n)$ to $r_\infty$.*

*Proof.* Because (A.1) is in the same form as the asynchronous update (Equation 7.1.2) by Borkar (2009), we apply the result in Section 7.4 of the same text (Borkar 2009) (see also Theorem 3.2 by Borkar (1998)) which shows convergence for Equation 7.1.2, to show the convergence of (A.1). This result, given Assumption A.6 and A.7, only requires showing the convergence of the following *synchronous* version of the General RVI Q algorithm:

$$Q_{n+1}(i) \doteq Q_n(i) + \alpha_n\big(R_n(i) - F_n(Q_n)(i) + g(Q_n)(i) - Q_n(i)\big) \quad \forall i \in \mathcal{I}. \tag{A.3}$$

Define operators $T_1, T_2$:

$$\begin{aligned} T_1(Q)(i) &\doteq r(i) + g(Q)(i) - r_\infty, \\ T_2(Q)(i) &\doteq r(i) + g(Q)(i) - f(Q) \\ &= T_1(Q)(i) + (r_\infty - f(Q)). \end{aligned}$$

Consider two ordinary differential equations (ODEs):

$$\dot{y}_t \doteq T_1(y_t) - y_t, \tag{A.4}$$
$$\dot{x}_t \doteq T_2(x_t) - x_t = T_1(x_t) - x_t + (r_\infty - f(x_t))\,e. \tag{A.5}$$

Note that because $g$ is a non-expansion by Assumption A.1, both (A.4) and (A.5) have Lipschitz R.H.S.'s and thus are well-posed.

Because $g$ is a non-expansion, $T_1$ is also a non-expansion. Therefore we have the next lemma, which restates Theorem 3.1 and Lemma 3.2 by Borkar and Soumyanath (1997).

**Lemma A.1.** *Let $\bar{y}$ be an equilibrium point of* (A.4). *Then $\|y_t - \bar{y}\|_\infty$ is nonincreasing, and $y_t \to y_*$ for some equilibrium point $y_*$ of* (A.4) *that may depend on $y_0$.*

**Lemma A.2.** (A.5) *has a unique equilibrium at $q_\infty$.*

*Proof.* Because $f(q_\infty) = r_\infty$, we have that $q_\infty = T_1(q_\infty) = T_2(q_\infty)$, thus $q_\infty$ is a equilibrium point for (A.5). Conversely, if $T_2(Q) - Q = 0$, then $T_1Q + (r_\infty - f(Q))e = Q$. But the equation $T_1Q + ce = Q$ only has a solution when $c = 0$ because of Assumption A.1. We have $c = 0$ and thus $f(Q) = r_\infty$, which along with $T_1Q = Q$, implies $Q = q_\infty$. $\square$

**Lemma A.3.** *Let $x_0 = y_0$, then $x_t = y_t + z_t e$, where $z_t$ satisfies the ODE $\dot{z}_t = -uz_t + (r_\infty - f(y_t))$, and $k \doteq |\mathcal{I}|$.*

*Proof.* From (A.4), (A.5), by the variation of parameters formula,

$$x_t = \exp(-t)x_0 + \int_0^t \exp(\tau - t)T_1(x_\tau)d\tau + \left[\int_0^t \exp(\tau - t)\left(r_\infty - f(x_\tau)\right)d\tau\right]e,$$

$$y_t = \exp(-t)y_0 + \int_0^t \exp(\tau - t)T_1(y_\tau)d\tau.$$

Then we have

$$\max_{s,o}(x_t(s,o) - y_t(s,o))$$

$$\leq \int_0^t \exp(\tau - t)\max_{s,o}(T_1(x_\tau)(s,o) - T_1(y_\tau)(s,o))d\tau + \left[\int_0^t \exp(\tau - t)\left(r_\infty - f(x_\tau)\right)d\tau\right],$$

$$\min_{s,o}(x_t(s,o) - y_t(s,o))$$

$$\geq \int_0^t \exp(\tau - t)\min_{s,o}(T_1(x_\tau)(s,o) - T_1(y_\tau)(s,o))d\tau + \left[\int_0^t \exp(\tau - t)\left(r_\infty - f(x_\tau)\right)d\tau\right].$$

Subtracting, we have

$$sp(x_t - y_t) \leq \int_0^t \exp(\tau - t)sp(T_1(x_\tau) - T_1(y_\tau))d\tau,$$

where $sp(x)$ denotes the span of vector $x$.

Because we assumed that $g$ is span-norm non-expansion, $T_1$ is also a span-norm non-expansion and thus

$$sp(x_t - y_t) \leq \int_0^t \exp(\tau - t)sp(T_1(x_\tau) - T_1(y_\tau))d\tau \leq \int_0^t \exp(\tau - t)sp(x_\tau - y_\tau)d\tau.$$

By Gronwall's inequality, $sp(x_t - y_t) = 0$ for all $t \geq 0$. Because $sp(x) = 0$ if and only if $x = ce$ for some $c \in \mathbb{R}$, we have

$$x_t = y_t + z_t e, \quad t \geq 0.$$

for some $z_t$. Also $x_0 = y_0 \implies z_0 = 0$.

Now we show that $\dot{z}_t = -uz_t + (r_\infty - f(y_t))$. Note that $f(x_t) = f(y_t + z_t e) = f(y_t) + uz_t$. In addition, $T_1(x_t) - T_1(y_t) = T_1(y_t + z_t e) - T_1(y_t) = T_1(y_t) + z_t e - T_1(y_t) = z_t e$, therefore we have, for $z_t \in \mathbb{R}$:

$$\begin{aligned}
\dot{z}_t e &= \dot{x}_t - \dot{y}_t \\
&= (T_1(x_t) - x_t + (r_\infty - f(x_t))\,e) - (T_1(y_t) - y_t) \quad \text{(from (A.4) and (A.5))} \\
&= -(x_t - y_t) + (T_1(x_t) - T_1(y_t)) + (r_\infty - f(x_t))\,e \\
&= -z_t e + z_t e + (r_\infty - f(x_t))\,e \\
&= -uz_t e + uz_t e + (r_\infty - f(x_t))\,e \\
&= -uz_t e + (r_\infty - f(y_t))\,e \\
\implies \dot{z}_t &= -uz_t + (r_\infty - f(y_t))\,.
\end{aligned}$$

$\square$

**Lemma A.4.** *$q_\infty$ is the globally asymptotically stable equilibrium for (A.5).*

*Proof.* We have shown that $q_\infty$ is the unique equilibrium in Lemma A.2.

With that result, we first prove Lyapunov stability. That is, we need to show that given any $\epsilon > 0$, we can find a $\delta > 0$ such that $\|q_\infty - x_0\|_\infty \leq \delta$ implies $\|q_\infty - x_t\|_\infty \leq \epsilon$ for $t \geq 0$.

First, from Lemma A.3 we have $\dot{z}_t = -uz_t + (r_\infty - f(y_t))$. By variation of parameters and $z_0 = 0$, we have

$$z_t = \int_0^t \exp(u(\tau - t)) \left(r_\infty - f(y_\tau)\right) d\tau.$$

Then

$$
\begin{aligned}
\|q_\infty - x_t\|_\infty &= \|q_\infty - y_t - z_t u e\|_\infty \\
&\leq \|q_\infty - y_t\|_\infty + u\,|z_t| \\
&\leq \|q_\infty - y_0\|_\infty + u \int_0^t \exp(u(\tau - t)) |r_\infty - f(y_\tau)| \, d\tau \\
&= \|q_\infty - x_0\|_\infty + u \int_0^t \exp(u(\tau - t)) |f(q_\infty) - f(y_\tau)| \, d\tau \quad \text{(from (A.2)).} \quad \text{(A.6)}
\end{aligned}
$$

Because $f$ is $L$-lipschitz, we have

$$
\begin{aligned}
|f(q_\infty) - f(y_\tau)| &\leq L\,\|r_\infty - y_\tau\|_\infty \\
&\leq L\,\|r_\infty - y_0\|_\infty \quad \text{(from Lemma A.1)} \\
&= L\,\|r_\infty - x_0\|_\infty .
\end{aligned}
$$

Therefore

$$
\begin{aligned}
\int_0^t \exp(u(\tau - t)) |f(q_\infty) - f(y_\tau)| \, d\tau &\leq \int_0^t \exp(u(\tau - t)) L\,\|q_\infty - x_0\|_\infty \, d\tau \\
&= L\,\|q_\infty - x_0\|_\infty \int_0^t \exp(u(\tau - t)) d\tau \\
&= L\,\|q_\infty - x_0\|_\infty \frac{1}{u}(1 - \exp(-ut)) \\
&= \frac{L}{u}\,\|q_\infty - x_0\|_\infty (1 - \exp(-ut)).
\end{aligned}
$$

Substituting the above equation in (A.6), we have

$$\|q_\infty - x_t\|_\infty \leq (1 + L)\,\|q_\infty - x_0\|_\infty .$$

Lyapunov stability follows.

Now in order to prove the asymptotic stability, in addition to Lyapunov stability, we need to show that there exists $\delta > 0$ such that if $\|x_0 - q_\infty\|_\infty < \delta$, then $\lim_{t \to \infty} \|x_t - q_\infty\|_\infty = 0$. Note that

$$
\begin{aligned}
\lim_{t \to \infty} z_t &= \lim_{t \to \infty} \int_0^t \exp(u(\tau - t)) \left(r_\infty - f(y_\tau)\right) d\tau \\
&= \lim_{t \to \infty} \frac{\int_0^t \exp(u\tau)(r_\infty - f(y_\tau)) d\tau}{\exp(ut)} \\
&= \lim_{t \to \infty} \frac{\exp(ut)(r_\infty - f(y_t))}{u \exp(ut)} \quad \text{(by L'Hospital's rule)} \\
&= \frac{r_\infty - f(y_\infty)}{u} \quad \text{(by Lemma A.1).}
\end{aligned}
$$

Because $x_t = y_t + z_t e$ (Lemma A.3) and $y_t \to y_\infty$ (Lemma A.1), we have $x_t \to y_\infty + (r_\infty - f(y_\infty))e/u$, which must coincide with $q_\infty$ because that is the only equilibrium point for (A.5) (Lemma A.2). Therefore $\lim_{t \to \infty} \|x_t - q_\infty\|_\infty = 0$ for any $x_0$. Asymptotic stability is shown and the proof is complete. $\qquad\square$

**Lemma A.5.** *Equation A.3 converges a.s. $Q_n$ to $q_\infty$ as $n \to \infty$.*

*Proof.* The proof uses Borkar's (2008) Theorem 2 in Section 2 and is essentially the same as Lemma 3.8 by Abounadi et al. (2001). For completeness, we repeat the proof (with more details) here.

First write the synchronous update (A.3) as

$$Q_{n+1} = Q_n + \alpha_n(h(Q_n) + M_{n+1} + \epsilon_n),$$

where

$$\begin{aligned}
h(Q_n)(i) &\doteq r(i) - f(Q_n) + g(Q_n)(i) - Q_n(i) \\
&= T_2(Q_n)(i) - Q_n(i), \\
M_{n+1}(i) &\doteq R_n(i) - F_n(Q_n)(i) + G_n(Q_n)(i) - T_2(Q_n)(i).
\end{aligned}$$

It can be shown that $\epsilon_n$ is asymptotically negligible and therefore does not affect the conclusions of Theorem 2 (text after Equation B.66 by Wan et al. 2021).

Theorem 2 requires verifying following conditions and concludes that $Q_n$ converges to a (possibly sample path dependent) compact connected internally chain transitive invariant set of ODE $\dot{x}_t = h(x_t)$. This is exactly the ODE defined in (A.5). Lemma A.2 and A.4 conclude that this ODE has $q_\infty$ as the unique globally asymptotically stable equilibrium. Therefore the (possibly sample path dependent) compact connected internally chain transitive invariant set is a singleton set containing only the unique globally asymptotically stable equilibrium. Thus Theorem 2 concludes that $Q_n \to q_\infty$ a.s. as $n \to \infty$. We now list conditions required by Theorem 2:

- **(A1)** The function $h$ is Lipschitz: $\|h(x) - h(y)\| \leq L\|x - y\|$ for some $0 < L < \infty$.

- **(A2)** The sequence $\{\alpha_n\}$ satisfies $\alpha_n > 0$, and $\sum \alpha_n = \infty$, $\sum \alpha_n^2 < \infty$.

- **(A3)** $\{M_n\}$ is a martingale difference sequence with respect to the increasing family of $\sigma$-fields

$$\mathcal{F}_n \doteq \sigma(Q_i, M_i, i \leq n), n \geq 0.$$

  That is

$$\mathbb{E}[M_{n+1} \mid \mathcal{F}_n] = 0 \quad \text{a.s.}, n \geq 0.$$

  Furthermore, $\{M_n\}$ are square-integrable

$$\mathbb{E}[\|M_{n+1}\|^2 \mid \mathcal{F}_n] \leq K(1 + \|Q_n\|^2) \quad \text{a.s.}, \quad n \geq 0,$$

  for some constant $K > 0$.

- **(A4)** $\sup_n \|Q_n\| \leq \infty$ a.s..

Let us verify these conditions now.

(A1) is satisfied because $T_2$ is Lipschitz.

(A2) is satisfied by Assumption A.5.

(A3) is also satisfied because for any $i \in \mathcal{I}$

$$\begin{aligned}
\mathbb{E}[M_{n+1}(i) \mid \mathcal{F}_n] &= \mathbb{E}\left[R_n(i) - F_n(Q_n)(i) + G_n(i) - T_2(Q_n)(i) \mid \mathcal{F}_n\right] \\
&= \mathbb{E}\left[R_n(i) - F_n(Q_n)(i) + G_n(Q_n)(i) \mid \mathcal{F}_n\right] - T_2(Q_n)(i) \\
&= 0,
\end{aligned}$$

and $\mathbb{E}[\|M_{n+1}\|^2 \mid \mathcal{F}_n] \leq \mathbb{E}[\|R_n - r\|^2 \mid \mathcal{F}_n] + \mathbb{E}[\|F_n(Q_n) - f(Q_n)e\|^2 \mid \mathcal{F}_n] + \mathbb{E}[\|G_n(Q_n) - g(Q_n)\|^2 \mid \mathcal{F}_n] \leq K(1 + \|Q_n\|^2)$ for a suitable constant $K > 0$ can be verified by a simple application of triangle inequality.

To verify (A4), we apply Theorem 7 in Section 3 by Borkar (2008), which shows $\sup_n \|Q_n\| \leq \infty$ a.s., if (A1), (A2), and (A3) are all satisfied and in addition we have the following condition satisfied:

**(A5)** The functions $h_d(x) \doteq h(dx)/d, d \geq 1, x \in \mathbb{R}^k$, satisfy $h_d(x) \to h_\infty(x)$ as $d \to \infty$, uniformly on compacts for some $h_\infty \in C(\mathbb{R}^k)$. Furthermore, the ODE $\dot{x}_t = h_\infty(x_t)$ has the origin as its unique globally asymptotically stable equilibrium.

Note that

$$h_\infty(x) = \lim_{d \to \infty} h_d(x) = \lim_{d \to \infty} (T_2(dx) - dx)/d = g(x) - f(x)e - x,$$

because $g(cx) = cg(x)$ and $f(cx) = cf(x)$ by our assumption.

The function $h_\infty$ is clearly continuous in every $x \in \mathbb{R}^k$ and therefore $h_\infty \in C(\mathbb{R}^k)$.

Now consider the ODE $\dot{x}_t = h_\infty(x_t) = g(x_t) - f(x_t)e - x_t$. Clearly the origin is an equilibrium. This ODE is a special case of (A.5), corresponding to the $r(s,o) \forall s \in \mathcal{S}, o \in \mathcal{O}$ being always zero. Therefore Lemma A.2 and A.4 also apply to this ODE and the origin is the unique globally asymptotically stable equilibrium.

(A1), (A2), (A3), (A4) are all verified and therefore

$$Q_n \to q_\infty \quad \text{a.s. as} \quad n \to \infty.$$

$\square$

$\square$

## A.2  Theorem 1

For simplicity, we will only provide formal theorems and proofs for our *control* learning and planning algorithms. The formal theorems and proofs for our prediction algorithms are similar to those for the control algorithms and are thus omitted. To this end, we first provide a general algorithm that includes both learning and planning control algorithms. We call it *General Inter-option Differential Q*. We first formally define it and then explain why both inter-option Differential Q-learning and inter-option Differential Q-planning are special cases of General Inter-option Differential Q. We then provide assumptions and the convergence theorem of the general algorithm. The theorem would lead to convergence of the special cases. Finally, we provide a proof for the theorem.

Given an SMDP $\hat{\mathcal{M}} = (\mathcal{S}, \mathcal{O}, \hat{\mathcal{L}}, \hat{\mathcal{R}}, \hat{p})$, for each state $s \in \mathcal{S}$, option $o \in \mathcal{O}$, and discrete step $n \geq 0$, let $\hat{R}_n(s,o), \hat{S}'_n(s,o), \hat{L}_n(s,o) \sim \hat{p}(\cdot, \cdot, \cdot | s, o)$ denote a sample of resulting state, reward and the length. We hypothesize a set-valued process $\{Y_n\}$ taking values in the set of nonempty subsets of $\mathcal{S} \times \mathcal{O}$ with the interpretation: $Y_n = \{(s,o) : (s,o) \text{ component of } Q \text{ was updated at time } n\}$. Let $\nu(n,s,o) \doteq \sum_{k=0}^n I\{(s,o) \in Y_k\}$, where $I$ is the indicator function. Thus $\nu(n,s,o) =$ the number of times the $(s,o)$ component was updated up to step $n$. The update rules of General Inter-option Differential Q are

$$Q_{n+1}(s,o) \doteq Q_n(s,o) + \alpha_{\nu(n,s,o)} \delta_n(s,o)/L_n(s,o)I\{(s,o) \in Y_n\}, \quad \forall s \in \mathcal{S}, o \in \mathcal{O}, \quad \text{(A.7)}$$

$$\bar{R}_{n+1} \doteq \bar{R}_n + \eta \sum_{s,o} \alpha_{\nu(n,s,o)} \delta_n(s,o)/L_n(s,o)I\{(s,o) \in Y_n\}, \quad \text{(A.8)}$$

$$L_{n+1}(s,o) \doteq L_n(s,o) + \beta_n(s,o)(\hat{L}_n(s,o) - L_n(s,o))I\{(s,o) \in Y_n\}, \quad \text{(A.9)}$$

where

$$\delta_n(s,o) \doteq \hat{R}_n(s,o) - \bar{R}_n L_n(s,o) + \max_{o'} Q_n(\hat{S}'_n(s,o), o') - Q_n(s,o) \quad \text{(A.10)}$$

is the TD error.

Here $\alpha_{\nu(n,s,o)}$ is the stepsize at step $n$ for state-action pair $(s,o)$. The quantity $\alpha_{\nu(n,s,o)}$ depends on the sequence $\{\alpha_n\}$, which is an algorithmic design choice, and also depends on the visitation of state-option pairs $\nu(n,s,o)$. To obtain the stepsize, the algorithm could maintain a $|\mathcal{S} \times \mathcal{O}|$-size table counting the number of visitations to each state-option pair, which is exactly $\nu(\cdot, \cdot, \cdot)$. Then the stepsize $\alpha_{\nu(n,s,o)}$ can be obtained as long as the sequence $\{\alpha_n\}$ is specified.

$Q_0$ and $R_0$ can be initialized arbitrarily. Note that $L_0$ can not be initialized to 0 because it is the divisor for both (A.7) and (A.8) for the first update. Because the expected length of all options would be greater than or equal to 1, we choose $L_0$ to be 1. In this way, $L_n$ will never be 0 because it is initialized to 1 and all the sampled option lengths are greater than or equal to 1. Therefore the problem of division by 0 will not happen in the updates.

Now we show inter-option Differential Q-learning and inter-option Differential Q-planning are special cases of General Inter-option Differential Q. Consider a sequence of real experience $\dots, \hat{S}_n, \hat{O}_n, \hat{R}_n, \hat{L}_n, \hat{S}_{n+1}, \dots$.

$$Y_n(s, o) = 1, \text{ if } s = \hat{S}_n, o = \hat{O}_n,$$
$$Y_n(s, o) = 0 \text{ otherwise,}$$

and $\hat{S}'_n(\hat{S}_n, \hat{O}_n) = \hat{S}_{n+1}, \hat{R}_n(\hat{S}_n, \hat{O}_n) = \hat{R}_{n+1}, \hat{L}_n(\hat{S}_n, \hat{O}_n) = \hat{L}_n$, update rules (A.7), (A.8), and (A.10) become

$$Q_{n+1}(\hat{S}_n, \hat{O}_n) \doteq Q_n(\hat{S}_n, \hat{O}_n) + \alpha_{\nu(n, \hat{S}_n, \hat{O}_n)} \hat{\delta}_n / L_n(\hat{S}_n, \hat{O}_n) \text{ , and } Q_{n+1}(s, o) \doteq Q_n(s, o), \forall s \neq \hat{S}_n, o \neq \hat{O}_n,$$
$$\bar{R}_{n+1} \doteq \bar{R}_n + \eta \alpha_{\nu(n, \hat{S}_n, \hat{O}_n)} \hat{\delta}_n / L_n(\hat{S}_n, \hat{O}_n),$$
$$\hat{\delta}_n \doteq \hat{R}_n - \bar{R}_n \hat{L}_n + \max_{o'} Q_n(\hat{S}_{n+1}, o') - Q_n(\hat{S}_n, \hat{O}_n),$$
$$L_{n+1}(\hat{S}_n, \hat{O}_n) \doteq L_n(\hat{S}_n, \hat{O}_n) + \beta_n(\hat{S}_n, \hat{O}_n)(\hat{L}_n - L_n(\hat{S}_n, \hat{O}_n))$$

which are inter-option Differential Q-learning's update rules (Section 3) with stepsize $\alpha$ in the $n$-th update being $\alpha_{\nu(n, \hat{S}_n, \hat{O}_n)}$, and the stepsize $\beta$ being $\beta(\hat{S}_n, \hat{O}_n)$.

If we consider a sequence of simulated experience $\dots, \tilde{S}_n, \tilde{O}_n, \tilde{R}_n, \tilde{L}_n, \tilde{S}'_n, \dots$.

$$Y_n(s, o) = 1, \text{ if } s = \tilde{S}_n, o = \tilde{O}_n,$$
$$Y_n(s, o) = 0 \text{ otherwise,}$$

and $\hat{S}'_n(s, o) = \tilde{S}'_n, \hat{R}_n(s, o) = \tilde{R}_n, \hat{L}_n(s, o) = \tilde{L}_n$, update rules (A.7)-(A.10) become

$$Q_{n+1}(\tilde{S}_n, \tilde{O}_n) \doteq Q_n(\tilde{S}_n, \tilde{O}_n) + \alpha_{\nu(n, \tilde{S}_n, \tilde{O}_n)} \tilde{\delta}_n / L_n \text{ , and } Q_{n+1}(s, o) \doteq Q_n(s, o), \forall s \neq \tilde{S}_n, o \neq \tilde{O}_n,$$
$$\bar{R}_{n+1} \doteq \bar{R}_n + \eta \alpha_{\nu(n, \tilde{S}_n, \tilde{O}_n)} \tilde{\delta}_n / L_n,$$
$$\tilde{\delta}_n \doteq \tilde{R}_n - \bar{R}_n \tilde{L}_n + \max_{o'} Q_n(\tilde{S}'_n, o') - Q_n(\tilde{S}_n, \tilde{O}_n),$$
$$L_{n+1}(\tilde{S}_n, \tilde{O}_n) \doteq L_n(\tilde{S}_n, \tilde{O}_n) + \beta_n(\tilde{S}_n, \tilde{O}_n)(\tilde{L}_n - L_n(\tilde{S}_n, \tilde{O}_n)).$$

Now, in the planning setting, the model can produce an expected length, instead of a sampled one. And there estimating the expected length using $L_n$ is no longer needed. The above updates reduce to

$$Q_{n+1}(\tilde{S}_n, \tilde{O}_n) \doteq Q_n(\tilde{S}_n, \tilde{O}_n) + \alpha_{\nu(n, \tilde{S}_n, \tilde{O}_n)} \tilde{\delta}_n / \tilde{L}_n \text{ , and } Q_{n+1}(s, o) \doteq Q_n(s, o), \forall s \neq \tilde{S}_n, o \neq \tilde{O}_n,$$
$$\bar{R}_{n+1} \doteq \bar{R}_n + \eta \alpha_{\nu(n, \tilde{S}_n, \tilde{O}_n)} \tilde{\delta}_n / \tilde{L}_n,$$
$$\tilde{\delta}_n \doteq \tilde{R}_n - \bar{R}_n \tilde{L}_n + \max_{o'} Q_n(\tilde{S}'_n, o') - Q_n(\tilde{S}_n, \tilde{O}_n).$$

The above update rules are our inter-option Differential Q-planning's update rules with stepsize at planning step $n$ being $\alpha_{\nu(n, \tilde{S}_n, \tilde{O}_n)}$.

We now provide a theorem, along with its proof, showing the convergence of General Inter-option Differential Q.

**Theorem A.2.** *Under Assumptions 1, A.5, A.6, A.7, and that $0 \leq \beta_n(s, o) \leq 1$, $\sum_n \beta_n(s, o) = \infty$, and $\sum_n \beta_n^2(s, o) < \infty$, and $\beta_n(s, o) = 0$ unless $s = \hat{S}_n$, General Inter-option Differential Q (Equations A.7-A.10) converges, almost surely, $Q_n$ to q satisfying both (2) and*

$$\eta\left(\sum q - \sum Q_0\right) = r_* - \bar{R}_0,$$

*$\bar{R}_n$ to $r_*$, and $r(\mu_n)$ to $r_*$ where $\mu_n$ is a greedy policy w.r.t. $Q_n$.*

*Proof.* At each step, the increment to $\bar{R}_n$ is $\eta$ times the increment to $Q_n$ and $\sum Q_n$. Therefore, the cumulative increment can be written

$$\bar{R}_n - \bar{R}_0 = \eta \sum_{i=0}^{n-1} \sum_{s,o} \alpha_{\nu(i,s,o)} \delta_i(s,o)/L_i(s,o) I\{(s,o) \in Y_i\}$$

$$= \eta \left( \sum Q_n - \sum Q_0 \right)$$

$$\implies \bar{R}_n = \eta \sum Q_n - \eta \sum Q_0 + \bar{R}_0 = \eta \sum Q_n - c, \tag{A.11}$$

$$\text{where } c \doteq \eta \sum Q_0 - \bar{R}_0. \tag{A.12}$$

Now substituting $\bar{R}_n$ in (A.7) with (A.11), we have $\forall s \in \mathcal{S}, o \in \mathcal{O}$:

$$Q_{n+1}(s,o) = Q_n(s,o) + \alpha_{\nu(n,s,o)}$$
$$\frac{\hat{R}_n(s,o) - L_n(s,o)(\eta \sum Q_n - c) + \max_{o'} Q_n(\hat{S}'_n(s,o),o') - Q_n(s,o)}{L_n(s,o)} I\{(s,o) \in Y_n\}$$

$$= Q_n(s,o) + \alpha_{\nu(n,s,o)}$$
$$\left( \frac{\hat{R}_n(s,o) - l_n(s,o)(\eta \sum Q_n - c) + \max_{o'} Q_n(\hat{S}'_n(s,o),o') - Q_n(s,o)}{l(s,o)} + \epsilon_n(s,o) \right) I\{(s,o) \in Y_n\},$$

$$\tag{A.13}$$

where $l(s,o)$ is the expected length of option $o$, starting from state $s$, and $\epsilon_n(s,o) \doteq (\hat{R}_n(s,o) - L_n(s,o)(\eta \sum Q_n - c) + \max_{o'} Q_n(\hat{S}'_n(s,o),o') - Q_n(s,o))/L(s,o) - (\hat{R}_n(s,o) - l(s,o)(\eta \sum Q_n - c) + \max_{o'} Q_n(\hat{S}'_n(s,o),o') - Q_n(s,o))/l(s,o)$.

Standard stochastic approximation result can be applied to show that $L_n$ converges to $l$. Further, it can be seen that $\epsilon_n$ satisfies that $\|\epsilon_n\|_\infty \leq K(1 + \|Q_n\|)$ for some positive $K$ and, by continuous mapping theorem, converges to 0 almost surely (and thus in probability).

We now show that (A.13) is a special case of (A.1). To see this point, let

$$i = (s,o),$$

$$R_n(i) = \frac{\hat{R}_n(s,o)}{l(s,o)} + c,$$

$$G_n(Q_n)(i) = \frac{\max_{o'} Q_n(\hat{S}'_n(s,o),o')}{l(s,o)} + \frac{l(s,o) - 1}{l(s,o)} Q_n(s,o),$$

$$F(Q_n)(i) = \eta \sum Q_n,$$

$$\epsilon_n(i) = \epsilon_n(s,o).$$

We now verify the assumptions of Theorem A.1 for Inter-option General Differential Q. Assumption A.1 and Assumption A.2 can be verified easily. Assumption A.3 satisfies because the MDP is finite. Assumption A.4 is satisfied as shown above. Assumption A.5-A.7 are satisfied due to assumptions of the theorem being proved. Assumption A.8 is satisfied because

$$r(i) - \bar{r} + g(q)(i) - q(i)$$
$$= \mathbb{E}[R_n(i) - \bar{r} + G_n(q)(i) - q(i)]$$
$$= \mathbb{E} \left[ \frac{\hat{R}_n(s,o) + cl(s,o) - \bar{r}l(s,o) + \max_{o'} q(\hat{S}'_n(s,o),o') + (l(s,o) - 1)q(s,o) - l(s,o)q(s,o)}{l(s,o)} \right]$$
$$= \frac{\mathbb{E} \left[ \hat{R}_n(s,o) + cl(s,o) - \bar{r}l(s,o) + \max_{o'} q(\hat{S}'_n(s,o),o') - q(s,o) \right]}{l(s,o)}.$$

From (2) we know if the above equation equals to 0, then under Assumption 1, $\bar{r} = r_* + c$ is the unique solution and the solutions for $q$ form a set $q = q_* + ce$.

All the assumptions are verified and thus from Theorem A.1 we conclude that $Q_n$ converges to a point satisfying $\eta \sum q = r_* + c = r_* + \eta \sum Q_0 - \bar{R}_0$ and $\bar{R}_n = \eta \sum Q_n - c$ to $\eta \sum q - c = r_* + c - c = r_*$.

Finally, in order to show $r(\mu_n) \to r_*$, we first extend Theorem 8.5.5 by Puterman (1994) to deal with SMDP.

**Lemma A.6.** *Under Assumption 1, $\forall Q \in \mathbb{R}^{|\mathcal{S} \times \mathcal{O}|}$*

$$\min_{s,o} TQ(s,o) \leq r(\mu_Q) \leq r_* \leq \max_{s,o} TQ(s,o),$$

*where $TQ(s,o) \doteq \sum_{s',r,l} \hat{p}(s',r,l \mid s,o)(r + \max_{o'} Q(s',o'))$ and $\mu_Q$ denotes a greedy policy w.r.t. $Q$.*

*Proof.* Note that

$$r(\mu_Q) = \sum_{s',r,l} \hat{p}(s',r,l \mid s,o)(r + \sum_{o'} \mu_Q(o' \mid s')Q(s',o') - Q(s,o)).$$

Therefore

$$\min_{s,o}(TQ_n(s,o) - Q_n(s,o)) \leq r(\mu_n) \leq r_* \leq \max_{s,o}(TQ_n(s,o) - Q_n(s,o))$$

$$\implies |r_* - r(\mu_n)| \leq sp(TQ_n - Q_n).$$

$\square$

Because $Q_n \to q_\infty$ a.s., and $sp(TQ_n - Q_n)$ is a continuous function of $Q_n$, by continuous mapping theorem, $sp(TQ_n - Q_n) \to sp(Tq_\infty - q_\infty) = 0$ a.s. Therefore we conclude that $r(\mu_n) \to r_*$.

$\square$

The convergence of General Inter-option Differential Q that we showed above implies Theorem 1 when there are no transient states ($\mathcal{S}' = \mathcal{S}$) and thus all states can be visited for an infinite number of times. When $\mathcal{S}' \subset \mathcal{S}$, option values associated states in $\mathcal{S} - \mathcal{S}'$ do not converge to a solution of the Bellman equation. However, the option values associated with recurrent states $\mathcal{S}'$ still converge to a solution of the Bellman equation, the reward rate estimator converges to $r_*$, and the $r(\mu_n)$ converges to $r_*$. The point that option values (associated with recurrent states) converge to depends on the sample trajectory. Specifically, it depends on the transient states visited in the trajectory.

### A.3 Theorem 2

The proof for the intra-option evaluation equation is simple. First note that these equations can be written in the vector form:

$$0 = r - \bar{r}e + (P_\mu - I)q,$$

where $r(s,o) = \mathbb{E}[R_{t+1} \mid S_t = s, O_t = o]$, $P_\mu((s,o),(s',o')) \doteq \Pr(S_{t+1} = s', O_{t+1} = o'|S_t = s, O_t = o, \mu) = \beta(s',o)\mu(o' \mid s') + (1 - \beta(s',o))\mathbb{I}(o = o')$, and $e$ is a all-one vector. Intuitively, the intra-option evaluation equation can be viewed as the evaluation equation for some average-reward MRP with reward and dynamics being defined as $r$ and $P_\mu$.

By Theorem 8.2.6 and Corollary 8.2.7 in Puterman (1994), the intra-option evaluation equation part in Theorem 2 is shown as long as the Markov chain associated with $P_\mu$ is unichain. Note that by our Assumption 1, there is only one recurrent class of states under any policy. Therefore no matter what the start state-option pair is, the agent will enter in the same recurrent class of states. Therefore we have, for every state $\bar{s}$ in the recurrent class and an option $\bar{o}$ such that $\mu(\bar{o} \mid \bar{s}) > 0$, the MDP visits $(\bar{s}, \bar{o})$ an an infinite number of times. This shows that any two state-option pairs can not be in two separate recurrent sets of state-option pairs. Therefore the Markov chain associated with $P_\mu$ is unichain.

The proof for the Intra-option Optimality Equations is more complicated. First, similar as what we know in the discounted primitive action case, we have the following lemma for the discounted option case.

**Lemma A.7.** *For every $0 < \gamma < 1$, there exists a stationary deterministic discount optimal policy.*

The proof uses similar arguments as Theorem 6.2.10 and Proposition 4.4.3 by Puterman (1994).

Now choose a sequence of discount factors $\{\gamma_n\}$, $0 \leq \gamma_n < 1$ with the property that $\gamma_n \uparrow 1$. By lemma A.7, for each $\gamma_n$, there exists a stationary discount optimal policy. Because the total number of Markov deterministic policies is finite, we can choose a subsequence $\{\gamma_n'\}$ for which the same Markov deterministic policy, $\mu$, is discount optimal for all $\gamma_n'$. Denote this subsequence by $\{\gamma_n\}$. Because $\mu$ is discount optimal for $\gamma_n, \forall n$, $q_*^{\gamma_n} = q_\mu^{\gamma_n}, \forall n$. By intra-option optimality equations in the discounted case (Sutton et al., 1999), for all $s \in \mathcal{S}, o \in \mathcal{O}$,

$$
\begin{aligned}
0 &= \sum_a \pi(a|s,o) \sum_{s',r} p(s',r|s,a) \left( r + \gamma_n \beta(s',o) q_\mu^{\gamma_n}(s',\mu(s')) + \gamma_n(1-\beta(s',o)) q_\mu^{\gamma_n}(s',o) \right) - q_\mu^{\gamma_n}(s,o) \\
&= \sum_a \pi(a|s,o) \sum_{s',r} p(s',r|s,a) \left( r + \gamma_n \beta(s',o) \max_{o'} q_\mu^{\gamma_n}(s',o') + \gamma_n(1-\beta(s',o)) q_\mu^{\gamma_n}(s',o) \right) - q_\mu^{\gamma_n}(s,o).
\end{aligned}
$$
$$(A.14)$$

By corollary 8.2.4 by Puterman (1994),

$$
q_\mu^{\gamma_n} = (1-\gamma_n)^{-1} r(\mu) e + q_\mu + f(\gamma_n), \tag{A.15}
$$

where $r(\mu)$ and $q_\mu$ are the reward rate and differential value function under policy $\mu$, and $f(\gamma)$ is a function of $\gamma$ that converges to 0 as $\gamma \uparrow 1$.

Substituting (A.15) into (A.14), we have

$$
\begin{aligned}
0 &= \sum_a \pi(a|s,o) \sum_{s',r} p(s',r|s,a)(r + \gamma_n \beta(s',o) \max_{o'}[(1-\gamma_n)^{-1} r(\mu) + q_\mu(s',o') + f(\gamma_n,s',o')] \\
&\quad + \gamma_n(1-\beta(s',o))[(1-\gamma_n)^{-1} r(\mu) + q_\mu(s',o) + f(\gamma_n,s',o)]) \\
&\quad - [(1-\gamma_n)^{-1} r(\mu) + q_\mu(s,o) + f(\gamma_n,s,o)] \\
&= \sum_a \pi(a|s,o) \sum_{s',r} p(s',r|s,a)(r - r(\mu) + \gamma_n \beta(s',o) \max_{o'}[q_\mu(s',o') + f(\gamma_n,s',o')] \\
&\quad + \gamma_n(1-\beta(s',o))[q_\mu(s',o) + f(\gamma_n,s',o)]) \\
&\quad - [q_\mu(s,o) + f(\gamma_n,s,o)] \\
&= \sum_a \pi(a|s,o) \sum_{s',r} p(s',r|s,a)(r - r(\mu) + \beta(s',o) \max_{o'}[q_\mu(s',o') + f(\gamma_n,s',o')] \\
&\quad + (\gamma_n - 1)\beta(s',o) \max_{o'}[q_\mu(s',o') + f(\gamma_n,s',o')] \\
&\quad + (1-\beta(s',o))[q_\mu(s',o) + f(\gamma_n,s',o)] \\
&\quad + (\gamma_n - 1)(1-\beta(s',o))[q_\mu(s',o) + f(\gamma_n,s',o)] \\
&\quad - [q_\mu(s,o) + f(\gamma_n,s,o)].
\end{aligned}
$$

Note that $(\gamma - 1)\beta(s',o) \max_{o'}[q_\mu(s',o') + f(\gamma,s',o')]$ and $(\gamma - 1)(1 - \beta(s',o))[q_\mu(s',o) + f(\gamma,s',o)]$ both converge to 0 as $\gamma \uparrow 1$.

Now take $n \to \infty$, then $\gamma_n \uparrow 1$, we have

$$
0 = \sum_a \pi(a|s,o) \sum_{s',r} p(s',r|s,a) \left( r - r(\mu) + \beta(s',o) \max_{o'} q_\mu(s',o') + (1-\beta(s',o)) q_\mu(s',o) \right) - q_\mu(s,o).
$$

We see that $\bar{r} = r(\mu)$ and $q = q_\mu$ is a solution of (11)-(12).

Now we show that the solution for $\bar{r}$ is unique. Define

$$
B(\bar{r},q) \doteq \sum_a \pi(a|s,o) \sum_{s',r} p(s',r|s,a) \left( r - \bar{r} + \beta(s',o) \max_{o'} q(s',o') + (1-\beta(s',o)) q(s',o) \right) - q(s,o).
$$

First we show if $B(\bar{r}, q) = 0$, then $\bar{r} \geq r_*$.

$$0 = B(\bar{r}, q)$$
$$= \sum_a \pi(a|s,o) \sum_{s',r} p(s',r|s,a)(r - \bar{r} + \beta(s',o) \max_{o'} q(s',o') + (1 - \beta(s',o))q(s',o)) - q(s,o)$$
$$\geq \sup_{\mu \in \Pi^{MR}} \sum_a \pi(a|s,o) \sum_{s',r} p(s',r|s,a)$$
$$\left( r - \bar{r} + \beta(s',o) \sum_{o'} \mu(o'|s')q(s',o') + (1 - \beta(s',o))q(s',o) \right) - q(s,o),$$

where $\Pi^{MR}$ denotes the set of all Markov randomized policies. In vector form, the above equation can be written as:

$$0 \geq \sup_{\mu \in \Pi^{MR}} \{ r - \bar{r}e + (P_\mu - I)q \}.$$

Therefore $\forall \mu \in \Pi^{MR}$,

$$\bar{r}e \geq r + (P_\mu - I)q.$$

Apply $P_\mu$ to both sides,

$$P_\mu \bar{r}e \geq P_\mu r + P_\mu (P_\mu - I)q,$$
$$\bar{r}e \geq P_\mu r + P_\mu (P_\mu - I)q.$$

Repeating this process we have:

$$\bar{r}e \geq P_\mu^n r + P_\mu^n (P_\mu - I)q.$$

Summing these expressions from $n = 0$ to $n = N - 1$ we have:

$$N\bar{r}e \geq \sum_{n=0}^{N-1} (P_\mu^n r + P_\mu^n (P_\mu - I)q) = \sum_{n=0}^{N-1} P_\mu^n r + (P_\mu^N - P_\mu^{N-1})q.$$

Because $\lim_{N \to \infty} \frac{1}{N}(P_\mu^N - P_\mu^{N-1})q = 0$,

$$\bar{r}e \geq \lim_{N \to \infty} \frac{1}{N} \sum_{n=0}^{N-1} P_\mu^n r = r(\mu)e,$$

for all $\mu \in \Pi^{MR}$. Therefore $\bar{r} \geq r_*$.

Then we show that if $0 = B(\bar{r}, q)$ then $\bar{r} \leq r_*$. As we proved above, if $(\bar{r}, q)$ satisfies that $0 = B(\bar{r}, q)$ then there exists a policy $\mu$ such that $\bar{r}e = r + (P_\mu - I)q$ is true. Therefore,

$$P_\mu^n \bar{r}e = P_\mu^n r + P_\mu^n (P_\mu - I)q,$$
$$\lim_{N \to \infty} \frac{1}{N} \sum_{n=0}^{N-1} P_\mu^n \bar{r}e = \lim_{N \to \infty} \frac{1}{N} \sum_{n=0}^{N-1} (P_\mu^n r + P_\mu^n (P_\mu - I)q),$$
$$\bar{r}e = \lim_{N \to \infty} \sum_{n=0}^{N-1} P_\mu^n r = r(\mu)e \leq r_* e.$$

Therefore $\bar{r} \leq r_*$. Combining $\bar{r} \geq r_*$ and $\bar{r} \leq r_*$ we have $\bar{r} = r_*$.

Finally, we show that the solution for $q$ is unique only up to a constant. Note that one could iteratively replace $q$ in the r.h.s. of the intra-option Optimality equation (11)-(12) by the entire r.h.s. of the intra-option Optimality equation, resulting to the inter-option Optimality equation (2). Therefore any solution of (11)-(12) must be a solution of (2). But we know that the solutions for $q$ in (2) is unique only up to a constant. Therefore the solutions for $q$ in (11)-(12) can not differ by a non-constant. Further, it is easy to see that if $q$ is a solution, then $q + ce, \forall c$ is also a solution. The theorem is proved.

$\square$

## A.4 Theorem 3

For simplicity, we will only provide formal theorems and proofs for our *control* learning and planning algorithms. The formal theorems and proofs for our prediction algorithms are similar to those for the control algorithms and are thus omitted. To this end, we first provide a general algorithm that includes both learning and planning control algorithms. We call it *General Intra-option Differential Q*. We first formally define it and then explain why both Intra-option Differential Q-learning and Intra-option Differential Q-planning are special cases of General Intra-option Differential Algorithm. We then provide assumptions and the convergence theorem of the general algorithm. The theorem would lead to convergence of the special cases. Finally, we provide a proof for the theorem.

Given an MDP $\mathcal{M} \doteq (\mathcal{S}, \mathcal{A}, \mathcal{R}, p)$ and a set of options $\mathcal{O}$, for each state $s \in \mathcal{S}$, option $o \in \mathcal{O}$, a reference option $\bar{o}$, and discrete step $n \geq 0$, let $A_n(s, \bar{o}) \sim \pi(\cdot \mid s, \bar{o})$, $R_n(s, A_n(s, \bar{o}))$, $S'_n(s, A_n(s, \bar{o})) \sim p(\cdot, \cdot \mid s, A_n(s, \bar{o}))$ denote, given state-option pair $(s, \bar{o})$, a sample of the chosen action and the resulting state and reward. We hypothesize a set-valued process $\{Y_n\}$ taking values in the set of nonempty subsets of $\mathcal{S} \times \mathcal{O}$ with the interpretation: $Y_n = \{(s, o) : (s, o) \text{ component of } Q \text{ was updated at time } n\}$. Let $\nu(n, s, o) \doteq \sum_{k=0}^{n} I\{(s, o) \in Y_k\}$, where $I$ is the indicator function. Thus $\nu(n, s, o) = $ the number of times the $(s, o)$ component was updated up to step $n$. In addition, we hypothesize a set-valued process $\{Z_n\}$ taking values in the set of nonempty subsets of $\mathcal{O}$ with the interpretation: $Z_n = \{\bar{o} : \bar{o} \text{ component was visited at time } n\}$. $\sum_{\bar{o}} I\{\bar{o} \in Z_n\}$ means the number of reference options used at update step $n$. For simplicity, we assume this number is always 1.

**Assumption A.9.** $\sum_{\bar{o}} I\{\bar{o} \in Z_n\} = 1$ *for all discrete $n \geq 0$.*

The update rules of General Intra-option Differential Q are

$$Q_{n+1}(s, o) \doteq Q_n(s, o) + \alpha_{\nu(n,s,o)} \sum_{\bar{o}} \rho_n(s, o, \bar{o}) \delta_n(s, o, \bar{o}) I\{(s, o) \in Y_n\} I\{\bar{o} \in Z_n\}, \quad \forall s \in \mathcal{S}, \text{ and } o \in \mathcal{O}$$

(A.16)

$$\bar{R}_{n+1} \doteq \bar{R}_n + \eta \sum_{s,o} \alpha_{\nu(n,s,o)} \sum_{\bar{o}} \rho_n(s, o, \bar{o}) \delta_n(s, o, \bar{o}) I\{(s, o) \in Y_n\} I\{\bar{o} \in Z_n\}, \quad \text{(A.17)}$$

where $\rho_n(s, o, \bar{o}) \doteq \pi(A_n(s, \bar{o}) \mid s, o) / \pi(A_n(s, \bar{o}) \mid s, \bar{o})$ and

$$\delta_n(s, o, \bar{o}) \doteq R_n(s, A_n(s, \bar{o})) - \bar{R}_n + \beta(S'_n(s, A_n(s, \bar{o})), o) \max_{o'} Q_n(S'_n(s, A_n(s, \bar{o})), o')$$

$$+ (1 - \beta(S'_n(s, A_n(s, \bar{o})), o)) Q_n(S'_n(s, A_n(s, \bar{o})), o) - Q_n(s, o) \quad \text{(A.18)}$$

is the TD error.

Here $\alpha_{\nu(n,s,o)}$ is the stepsize at step $n$ for state-option-option triple $(s, o)$. The quantity $\alpha_{\nu(n,s,o)}$ depends on the sequence $\{\alpha_n\}$, which is an algorithmic design choice, and also depends on the visitation of state-option pairs $\nu(n, s, o)$. To obtain the stepsize, the algorithm could maintain a $|\mathcal{S} \times \mathcal{O}|$-size table counting the number of visitations to each state-option pair, which is exactly $\nu(\cdot, \cdot, \cdot)$. Then the stepsize $\alpha_{\nu(n,s,o)}$ can be obtained as long as the sequence $\{\alpha_n\}$ is specified.

Now we show Intra-option Differential Q-learning and Intra-option Differential Q-planning are special cases of General Intra-option Differential Q. Consider a sequence of real experience $\ldots, S_t, O_t, A_t, R_{t+1}, S_{t+1}, \ldots$. By choosing step $n = $ time step $t$,

$$Y_n(s, o) = 1, \text{ if } s = S_t$$
$$Y_n(s, o) = 0 \text{ otherwise,}$$
$$Z_n(\bar{o}) = 1, \text{ if } \bar{o} = O_t$$
$$Z_n(\bar{o}) = 0 \text{ otherwise,}$$

and $A_n(S_t, O_t) = A_t$, $S'_n(S_t, A_n(S_t, O_t)) = S_{t+1}$, $R_n(S_t, A_n(S_t, O_t)) = R_{t+1}$, update rules (A.16), (A.17), and (A.18) become

$$Q_{t+1}(S_t, o) \doteq Q_t(S_t, o) + \alpha_{\nu(t,S_t,o)} \rho_t(o) \delta_t(o), \forall o \in \mathcal{O}, \text{ and } Q_{t+1}(s, o) \doteq Q_t(s, o), \forall o \in \mathcal{O} \text{ and } \forall s \neq S_t,$$

$$\bar{R}_{t+1} \doteq \bar{R}_t + \eta \sum_o \alpha_{\nu(t,S_t,o)} \rho_t(o) \delta_t(o),$$

$$\delta_t(o) \doteq R_{t+1} - \bar{R}_t + \beta(S_{t+1}, o) \max_{o'} Q_t(S_{t+1}, o') + (1 - \beta(S_{t+1}, o)) Q_t(S_{t+1}, o) - Q_t(S_t, o),$$

where $\rho_t(o) \doteq \pi(A_t \mid S_t, o)/\pi(A_t \mid S_t, O_t)$. The above equations are Intra-option Differential Q-learning's update rules (Equations 14, 15, 16) with stepsize at time $t$ being $\alpha_{\nu(t,S_t,o)}$ for each option $o$.

If we consider a sequence of simulated experience $\ldots, \tilde{S}_n, \tilde{O}_n, \tilde{A}_n, \tilde{R}_n, \tilde{S}'_n, \ldots$, by choosing step $n = $ planning step $n$,

$$Y_n(s, o) = 1, \text{ if } s = \tilde{S}_n$$
$$Y_n(s, o) = 0 \text{ otherwise,}$$
$$Z_n(\bar{o}) = 1, \text{ if } \bar{o} = \tilde{O}_n$$
$$Z_n(\bar{o}) = 0 \text{ otherwise,}$$

and $A_n(\tilde{S}_n, \tilde{O}_n) = \tilde{A}_n$, $S'_n(\tilde{S}_n, A_n(\tilde{S}_n, \tilde{O}_n)) = \tilde{S}'_n$, $R_n(\tilde{S}_n, A_n(\tilde{S}_n, \tilde{O}_n)) = \tilde{R}_n$, update rules (A.16), (A.17), and (A.18) become

$$Q_{n+1}(\tilde{S}_n, o) \doteq Q_n(\tilde{S}_n, o) + \alpha_{\nu(n,\tilde{S}_n,o)}\rho_n(o)\delta_n(o), \forall o \in \mathcal{O} \text{ , and } Q_{n+1}(s, o) \doteq Q_n(s, o), \forall s \neq \tilde{S}_n, \forall o \in \mathcal{O}$$

$$\bar{R}_{n+1} \doteq \bar{R}_n + \eta \sum_o \alpha_{\nu(n,\tilde{S}_n,o)}\rho_n(o)\delta_n(o),$$

$$\delta_n(o) \doteq \tilde{R}_n - \bar{R}_n + \beta(\tilde{S}'_n, o) \max_{o'} Q_n(\tilde{S}'_n, o') + (1 - \beta(\tilde{S}'_n, o))Q_n(\tilde{S}'_n, o) - Q_n(\tilde{S}_n, o),$$

where $\rho_n(o) \doteq \pi(A_n \mid S_n, o)/\pi(A_n \mid S_n, O_n)$. The above equations are Intra-option Differential Q-planning's update rules (Equations 14, 15, 16) with stepsize at planning step $n$ being $\alpha_{\nu(n,S_n,o)}$ for each option $o$.

Finally, note that for both Intra-option Differential Q-learning and Q-planning algorithms, because for each time step $t$ or update step $n$, there is only one option which is actually chosen to generate data, Assumption A.9 is satisfied.

**Theorem A.3.** *Under Assumptions 1, A.5, A.6, A.7, A.9, General Intra-option Differential Q (Equations A.16-A.18) converges, almost surely, $Q_n$ to q satisfying both* (11)-(12) *and*

$$\eta(\sum q - \sum Q_0) = r_* - \bar{R}_0, \tag{A.19}$$

$\bar{R}_n$ *to* $r_*$, *and* $r(\mu_n)$ *to* $r_*$ *where* $\mu_n$ *is a greedy policy w.r.t.* $Q_n$.

*Proof.* At each step, the increment to $\bar{R}_n$ is $\eta$ times the increment to $Q_n$ and $\sum Q_n$. Therefore, the cumulative increment can be written as:

$$\bar{R}_n - \bar{R}_0 = \eta \sum_{i=0}^{n-1} \sum_{s,o} \alpha_{\nu(i,s,o)} \sum_{\bar{o}} \rho_i(s, o, \bar{o})\delta_i(s, o, \bar{o})I\{(s, o) \in Y_i\}I\{\bar{o} \in Z_i\}$$

$$= \eta\left(\sum Q_n - \sum Q_0\right)$$

$$\implies \bar{R}_n = \eta\sum Q_n - \eta\sum Q_0 + \bar{R}_0 = \eta\sum Q_n - c, \tag{A.20}$$

$$\text{where } c \doteq \eta \sum Q_0 - \bar{R}_0.$$

Now substituting $\bar{R}_n$ in (A.16) with (A.20), we have $\forall s \in \mathcal{S}, o \in \mathcal{O}$:

$$Q_{n+1}(s, o) = Q_n(s, o) + \alpha_{\nu(n,s,o)} \sum_{\bar{o}} \frac{\pi(A_n(s, \bar{o}) \mid s, o)}{\pi(A_n(s, \bar{o}) \mid s, \bar{o})}$$

$$\left(R_n(s, A_n(s, \bar{o})) - \eta\sum Q_n + c + \beta(S'_n(s, A_n(s, \bar{o})), o) \max_{o'} Q_n(S'_n(s, A_n(s, \bar{o})), o')\right.$$

$$\left. + (1 - \beta(S'_n(s, A_n(s, \bar{o})), o))Q_n(S'_n(s, A_n(s, \bar{o})), o) - Q_n(s, o)\right)$$

$$I\{(s, o) \in Y_n\}I\{\bar{o} \in Z_n\}. \tag{A.21}$$

We now show that (A.21) is a special case of (A.1). To see this point, let $i = (s, o)$,

$$R_n(i) = \sum_{\bar{o}} \frac{\pi(A_n(s, \bar{o}) \mid s, o)}{\pi(A_n(s, \bar{o}) \mid s, \bar{o})} I\{\bar{o} \in Z_n\}(R_n(s, A_n(s, \bar{o})) + c),$$

$$F_n(Q_n)(i) = \sum_{\bar{o}} \frac{\pi(A_n(s, \bar{o}) \mid s, o)}{\pi(A_n(s, \bar{o}) \mid s, \bar{o})} I\{\bar{o} \in Z_n\} \eta \sum Q_n,$$

$$G_n(Q_n)(i) = \sum_{\bar{o}} \frac{\pi(A_n(s, \bar{o}) \mid s, o)}{\pi(A_n(s, \bar{o}) \mid s, \bar{o})} I\{\bar{o} \in Z_n\}\big(\beta(S_n'(s, A_n(s, \bar{o})), o) \max_{o'} Q_n(S_n'(s, A_n(s, \bar{o})), o')$$
$$+ (1 - \beta(S_n'(s, A_n(s, \bar{o})), o))Q_n(S_n'(s, A_n(s, \bar{o})), o) - Q_n(s, o)\big),$$

$$\epsilon_n(i) = 0.$$

Then we have:

$$r(i) = \mathbb{E}[R_n(i)]$$

$$= \mathbb{E}\left[\sum_{\bar{o}} \frac{\pi(A_n(s, \bar{o}) \mid s, o)}{\pi(A_n(s, \bar{o}) \mid s, \bar{o})} I\{\bar{o} \in Z_n\}(R_n(s, A_n(s, \bar{o})) + c)\right]$$

$$= \sum_{\bar{o}} \mathbb{E}\left[\frac{\pi(A_n(s, \bar{o}) \mid s, o)}{\pi(A_n(s, \bar{o}) \mid s, \bar{o})} I\{\bar{o} \in Z_n\}(R_n(s, A_n(s, \bar{o})) + c)\right]$$

$$= \sum_{\bar{o}} I\{\bar{o} \in Z_n\} \sum_a \pi(a \mid s, o)\mathbb{E}[R_n(s, a) + c]$$

$$= \sum_a \pi(a \mid s, o) \sum_{r, s'} p(r, s' \mid s, a)(r + c), \qquad \text{By Assumption A.9,}$$

$$f(q) = \mathbb{E}[F(q)(i)] = \eta \sum q,$$

$$g(q)(i) = \mathbb{E}[G_n(q)(i)]$$

$$= \mathbb{E}\left[\sum_{\bar{o}} \frac{\pi(A_n(s, \bar{o}) \mid s, o)}{\pi(A_n(s, \bar{o}) \mid s, \bar{o})} I\{\bar{o} \in Z_n\}\big(\beta(S_n'(s, A_n(s, \bar{o})), o) \max_{o'} q(S_n'(s, A_n(s, \bar{o})), o')\right.$$

$$\left. + (1 - \beta(S_n'(s, A_n(s, \bar{o})), o))q(S_n'(s, A_n(s, \bar{o})), o) - q(s, o)\big)\right]$$

$$= \sum_{\bar{o}} I\{\bar{o} \in Z_n\} \sum_a \pi(a \mid s, o)$$

$$\mathbb{E}[(\beta(S_n'(s, a), o) \max_{o'} q(S_n'(s, a), o') + (1 - \beta(S_n'(s, a), o))q(S_n'(s, a), o) - q(s, o))]$$

$$= \sum_a \pi(a \mid s, o) \sum_{s', r} p(s', r \mid s, a)(\beta(s', o) \max_{o'} q(s', o') + (1 - \beta(s', o))q(s', o) - q(s, o))],$$

for any $i \in \mathcal{I}$.

We now verify the assumptions of Theorem A.1 for Intra-option General Differential Q. Assumption A.1 can be verified for $g(q)(s, o) = \sum_a \pi(a \mid s, o) \sum_{s', r} p(s', r \mid s, a)(\beta(s', o) \max_{o'} q(s', o') + (1 - \beta(s', o))q(s', o))$ easily. Assumption A.2 is satisfied for $f(q) = \eta \sum q$. Assumption A.3 satisfies because the MDP is finite. Assumption A.4 is satisfied for $\epsilon_n = 0$. Assumption A.5-A.7 are satisfied due to assumptions of the theorem being proved. Assumption A.8 is satisfied because

$$r(i) - \bar{r} + g(q)(i) - q(i)$$

$$= \sum_a \pi(a|s, o) \sum_{s', r} p(s', r|s, a)(r - \bar{r} + \beta(s', o) \max_{o'} q(s', o') + (1 - \beta(s', o))q(s', o)).$$

By Theorem 2, we know that if the above equation equals to 0, then under Assumption 1, $\bar{r} = r_* + c$ is the unique solution and the solutions for $q$ form a set $q = q_* + ke$ for all $k \in \mathbb{R}$.

Therefore Theorem A.1 applies and we conclude that $Q_n$ converges to a point satisfying $\eta \sum q = r_* + c = r_* + \eta \sum Q_0 - R_0$ and $\bar{R}_n = \eta \sum Q_n - c$ to $\eta \sum q - c = r_* + c - c = r_*$. Finally, by Lemma A.6, we have $r(\mu_n) \to r_*$.

Applying a similar argument as one presented in the last paragraph of Section A.2 finishes the proof of Theorem 3.

$\square$

## A.5 Theorem 4

*Proof.* We will show that there exists a unique solution for (18). Results for (19) and (20) can be shown in a similar way. To show that (18) has a unique solution, we apply a generalized version of the Banach fixed point theorem (see, e.g., Theorem 2.4 by Almezel, Ansari, and Khamsi 2014). Once the unique existence of the solution is shown, we easily verify that $m^p$ is the unique solution by showing that it is one solution to (18) as follows. With a little abuse of notation, let $\hat{p}(s', r \mid s, o) \doteq \sum_{r,l} \hat{p}(x, r, l \mid s, o)$, we have

$$
\begin{aligned}
m^p(x|s,o) &= \sum_{r,l} \hat{p}(x,r,l|s,o) \\
&= \sum_{l=1}^{\infty} \hat{p}(x,l|s,o) = \sum_a \pi(a|s,o) \sum_r p(s',r|s,a)\beta(s',o)\mathbb{I}(x=s') + \sum_{l=2}^{\infty} \hat{p}(x,l|s,o) \\
&= \sum_a \pi(a|s,o) \sum_r p(s',r|s,a)\big(\beta(s',o)\mathbb{I}(x=s') + (1-\beta(s',o))\sum_{l=1}^{\infty} \hat{p}(x,l|s',o)\big) \\
&= \sum_a \pi(a|s,o) \sum_r p(s',r|s,a)\big(\beta(s',o)\mathbb{I}(x=s') + (1-\beta(s',o))m^p(x|s',o)\big).
\end{aligned}
$$

To apply the generalized version of the Banach fixed point theorem to show the unique existence of the solution, we first define operator $T : \mathbb{R}^{|\mathcal{S}|\times|\mathcal{S}|\times|\mathcal{O}|} \rightarrow \mathbb{R}^{|\mathcal{S}|\times|\mathcal{S}|\times|\mathcal{O}|}$ such that for any $m \in \mathbb{R}^{|\mathcal{S}|\times|\mathcal{S}|\times|\mathcal{O}|}$ and any $x,s \in \mathcal{S}, o \in \mathcal{O}$, $Tm(x \mid s,o) \doteq \sum_a \pi(a|s,o) \sum_{s',r} p(s',r|s,a)(\beta(s',o)\mathbb{I}(x=s') + (1-\beta(s',o))m(x|s',o)))$. We further define $T^n m \doteq T(T^{n-1}m)$ for any $n \geq 2$ and any $m \in \mathbb{R}^{|\mathcal{S}|\times|\mathcal{S}|\times|\mathcal{O}|}$. The generalized Banach fixed point theorem shows that if $T^n$ is a contraction mapping for any integer $n \geq 1$ (this is called a $n$-stage contraction), then $Tm = m$ has a unique fixed point. The unique fixed point immediately leads to the existence of the unique solution of (18). The existence of the unique solution and that $m^p$ is a solution imply that $m^p$ is the unique solution.

The only work left is to verify the following contraction property:

$$
\left\| T^{|\mathcal{S}|}m_1 - T^{|\mathcal{S}|}m_2 \right\|_\infty \leq \gamma \left\| m_1 - m_2 \right\|_\infty, \tag{A.22}
$$

where $m_1$ and $m_2$ are arbitrary members in $\mathbb{R}^{|\mathcal{S}|\times|\mathcal{S}|\times|\mathcal{O}|}$, and $\gamma < 1$ is some constant.

Consider the difference between $T^{|\mathcal{S}|}m_1$ and $T^{|\mathcal{S}|}m_2$ for arbitrary $m_1, m_2 \in \mathbb{R}^{|\mathcal{S} \times \mathcal{S} \times \mathcal{O}|}$. For any $x, s \in \mathcal{S}, o \in \mathcal{O}$, we have

$$T^{|\mathcal{S}|}m_1(x \mid s, o) - T^{|\mathcal{S}|}m_2(x \mid s, o)$$

$$= \sum_a \pi(a \mid s, o) \sum_{s', r} p(s', r \mid s, a)(1 - \beta(s', o))(T^{|\mathcal{S}|-1}m_1(x \mid s', o) - T^{|\mathcal{S}|-1}m_2(x \mid s', o))$$

$$= \sum_{s_1} \Pr(S_{t+1} = s_1 \mid S_t = s, O_t = o)(1 - \beta(s_1, o))(T^{|\mathcal{S}|-1}m_1(x \mid s_1, o) - T^{|\mathcal{S}|-1}m_2(x \mid s_1, o))$$

$$= \sum_{s_1} \Pr(S_{t+1} = s_1 \mid S_t = s, O_t = o)(1 - \beta(s_1, o)) \sum_{s_2} \Pr(S_{t+2} = s_2 \mid S_{t+1} = s_1, O_{t+1} = o)(1 - \beta(s_2, o))$$

$$(T^{|\mathcal{S}|-2}m_1(x \mid s_2, o) - T^{|\mathcal{S}|-2}m_2(x \mid s_2, o))$$

$$\vdots$$

$$= \sum_{s_1, \cdots, s_{|\mathcal{S}|}} \Pr(S_{t+1} = s_1, \cdots, S_{t+|\mathcal{S}|} = s_{|\mathcal{S}|} \mid S_t = s, O_t = o) \prod_{i=1}^{|\mathcal{S}|} (1 - \beta(s_i, o))(m_1(x \mid s_{|\mathcal{S}|}, o) - m_2(x \mid s_{|\mathcal{S}|}, o))$$

$$\leq \sum_{s_1, \cdots, s_{|\mathcal{S}|}} \Pr(S_{t+1} = s_1, \cdots, S_{t+|\mathcal{S}|} = s_{|\mathcal{S}|} \mid S_t = s, O_t = o) \prod_{i=1}^{|\mathcal{S}|} (1 - \beta(s_i, o)) \|m_1 - m_2\|_\infty.$$

Here $\tilde{p}(s, o) \doteq \sum_{s_1, \cdots, s_{|\mathcal{S}|}} \Pr(S_{t+1} = s_1 \cdots, S_{t+|\mathcal{S}|} = s_{|\mathcal{S}|} \mid S_t = s, O_t = o) \prod_{i=1}^{|\mathcal{S}|} (1 - \beta(s_i, o))$ is the probability of executing option $o$ for $|\mathcal{S}|$ steps starting from $s$ without termination. If $\tilde{p}(s, o) = 0$, then option $o$ will surely terminate within the first $|\mathcal{S}|$ steps and if $\tilde{p}(s, o) = 1$, then option $o$ will surely not terminate within the first $|\mathcal{S}|$ steps.

If option $o$ would surely not terminate within the first $|\mathcal{S}|$ steps ($\tilde{p}(s, o) = 1$), then it would surely not terminate forever. This is because there are only $|\mathcal{S}|$ number of states, and thus an option could visit all states that are possible to be visited by the option within the first $|\mathcal{S}|$ steps. $\tilde{p}(s, o) = 1$ means that option $o$ has a zero probability of terminating in all states that are possible to be visited by option $o$. This non-termination of a state-option pair implies that the expected option length is infinite, which is contradict to our assumption of finite expected option lengths (Section 2). Therefore $\tilde{p}(s, o) = 1$ is not allowed by our assumption and thus $\tilde{p}(s, o) < 1$. So there must exist some $\gamma(s, o) < 1$ such that $\tilde{p}(s, o) \leq \gamma(s, o)$. With $\gamma \doteq \max_{s,o} \gamma(s, o)$, we obtain (A.22). $\qquad\square$

## A.6   Theorem 5

We first provide a formal statement of Theorem 5. The formal theorem statement needs stepsizes to be specific for each state-option pair. We rewrite (21–23) to incorporate such stepsizes:

$$M_{t+1}^p(x \mid S_t, o) \doteq M_t^p(x \mid S_t, o) + \alpha_t(S_t, o)\rho_t(o)\Big(\beta(S_{t+1}, o)\mathbb{I}(S_{t+1} = x)$$

$$+ \big(1 - \beta(S_{t+1}, o)\big)M_t^p(x \mid S_{t+1}, o) - M_t^p(x \mid S_t, o)\Big), \quad \forall\, x \in \mathcal{S}, \tag{A.23}$$

$$M_{t+1}^r(S_t, o) \doteq M_t^r(S_t, o) + \alpha_t(S_t, o)\rho_t(o)\Big(R_{t+1} + \big(1 - \beta(S_{t+1}, o)\big)M_t^r(S_{t+1}, o) - M_t^r(S_t, o)\Big) \tag{A.24}$$

$$M_{t+1}^l(S_t, o) \doteq M_t^l(S_t, o) + \alpha_t(S_t, o)\rho_t(o)\Big(1 + \big(1 - \beta(S_{t+1}, o)\big)M_t^l(S_{t+1}, o) - M_t^l(S_t, o)\Big). \tag{A.25}$$

**Theorem A.4** (Convergence of the intra-option model learning algorithm, formal)**.** *If* $0 \leq \alpha_t(s, o) \leq 1$, $\sum_t \alpha_t(s, o) = \infty$ *and* $\sum_t \alpha_t^2(s, o) < \infty$, *and* $\alpha_t(s, o) = 0$ *unless* $s = S_t$, *then the intra-option model-learning algorithm (A.23–A.25) converges almost surely,* $M_t^p$ *to* $m^p$, $M_t^r$ *to* $m^r$, *and* $M_t^l$ *to* $m^l$.

Here the assumptions on $\alpha_t$ guarantee that each state-option pair is updated for an infinite number of times. Because the three update rules are independent, we only show convergence of the first update rule; the other two can be shown in the same way.

*Proof.* We apply a slight generalization of Theorem 3 by Tsitsiklis (1994) to show the above theorem. The generalization replaces Assumption 5 (an assumption for Theorem 3) by:

**Assumption A.10.** *There exists a vector $x^* \in \mathbb{R}^n$, a positive vector $v$, a positive integer $m$, and a scalar $\beta \in [0, 1)$, such that*

$$\|F^m(x) - x^*\|_v \leq \beta \|x - x^*\|_v, \quad \forall x \in \mathbb{R}^n.$$

That is, we replace the one-stage contraction assumption by a $m$-stage contraction assumption. The proof of Tsitsiklis' Theorem 3 also applies to its generalized form and is thus omitted here.

Notice that our update rule (A.23) is a special case of the general update rule considered by Theorem 3 (equations 1-3), and is thus a special case of its generalized version. Therefore we only need to verify the above $m-$stage contraction assumption, as well as Assumption 1, 2, and 3 required by Theorem 3. According to the proof in Appendix A.5, the operator $T$ associated with the update rule (21) is a $|\mathcal{S}|-$stage contraction (and thus is a $|\mathcal{S}|-$stage pseudo-contraction). Other assumptions (Assumptions 1, 2, 3) required by Theorem 3 are also satisfied given our step-size, and finite MDP assumptions. $\qquad\square$

### A.7 Theorem 6

*Proof.* We first show that

$$\sum_{o'} \mu'(o' \mid s) \sum_{s',r,l} \hat{p}(s',r,l \mid s,o')(r - lr(\mu) + v_\mu(s'))$$

$$\geq \sum_{o} \mu(o \mid s) \sum_{s',r,l} \hat{p}(s',r,l \mid s,o)(r - lr(\mu) + v_\mu(s')) = v_\mu(s). \tag{A.26}$$

Note that for all $s$, $o$ and its corresponding $o'$, $\mu(o \mid s) = \mu'(o' \mid s)$. In order to show (A.26), we show $\sum_{s',r,l} \hat{p}(s',r,l \mid s,o')(r - lr(\mu) + v_\mu(s')) \geq \sum_{s',r,l} \hat{p}(s',r,l \mid s,o)(r - lr(\mu) + v_\mu(s'))$ for all $s$, $o$ and corresponding $o'$.

$$\sum_{s',r,l} \hat{p}(s',r,l \mid s,o')(r - lr(\mu) + v_\mu(s'))$$

$$= \mathbb{E}[\hat{R}_n - \hat{L}_n r(\mu) + v_\mu(\hat{S}_{n+1}) \mid S_n = s, O_n = o']$$

$$= \mathbb{E}[\hat{R}_n - \hat{L}_n r(\mu) + v_\mu(\hat{S}_{n+1}) \mid S_n = s, O_n = o', \text{Not encountering an interruption}]$$

$$+ \mathbb{E}[\hat{R}_n - \hat{L}_n r(\mu) + v_\mu(\hat{S}_{n+1}) \mid S_n = s, O_n = o', \text{Encountering an interruption}]$$

$$\geq \mathbb{E}[\hat{R}_n - \hat{L}_n r(\mu) + v_\mu(\hat{S}_{n+1}) \mid S_n = s, O_n = o', \text{Not encountering an interruption}]$$

$$+ \mathbb{E}[\beta(s')(\hat{R}_n - \hat{L}_n r(\mu) + v_\mu(\hat{S}_{n+1})) + (1 - \beta(s'))(\hat{R}_n - \hat{L}_n r(\mu) + q_\mu(\hat{S}_{n+1}, o))$$

$$\mid S_n = s, O_n = o', \text{Encountering an interruption}]$$

$$= \sum_{s',r,l} \hat{p}(s',r,l \mid s,o)(r - lr(\mu) + v_\mu(s')).$$

The above inequality holds because $\hat{S}_{n+1}$ is the state where termination happens and thus $q_\mu(\hat{S}_{n+1}, o) \leq v_\mu(\hat{S}_{n+1})$. The last equality holds because $\mathbb{E}[\beta(s')(\hat{R}_n - \hat{L}_n r(\mu) + v_\mu(\hat{S}_{n+1})) + (1 - \beta(s'))(\hat{R}_n - \hat{L}_n r(\mu) + q_\mu(\hat{S}_{n+1}, o)) \mid S_n = s, O_n = o', \text{Encountering an interruption}]$ is the expected differential return when the agent could interrupt its old option but chooses to stick on the old option. (A.26) is shown.

Now write the l.h.s. of (A.26) in the matrix form

$$\sum_{o'} \mu'(o' \mid s) \sum_{s',r,l} \hat{p}(s',r,l|s,o')(r - lr(\mu) + v_\mu(s')) = r_{\mu'}(s) - l_{\mu'}(s)r(\mu) + (P_{\mu'}v_\mu)(s),$$

where $r_{\mu'}(s) \doteq \sum_{o'} \mu'(o' \mid s) \sum_{s',r,l} \hat{p}(s',r,l|s,o')r$ is the expected one option-transition reward, $l_{\mu'}(s) \doteq \sum_{o'} \mu'(o' \mid s) \sum_{s',r,l} \hat{p}(s',r,l|s,o')l$ is the expected one option-transition length, and $P_{\mu'}(s,s') \doteq \sum_{o'} \mu'(o' \mid s) \sum_{r,l} \hat{p}(s',r,l|s,o')$ is the probability of terminating at $s'$.

Combined with the r.h.s. of (A.26), we have

$$r_{\mu'}(s) - l_{\mu'}(s)r(\mu) + (P_{\mu'}v_\mu)(s) \geq v_\mu(s).$$

Iterating the above inequality for $K - 1$ times, we have

$$\sum_{k=0}^{K-1}(P_{\mu'}^k r_{\mu'}(s) - P_{\mu'}^k l_{\mu'}(s)r(\mu)) + P_{\mu'}^K v_\mu(s) \geq v_\mu(s)$$

$$\sum_{k=0}^{K-1}(P_{\mu'}^k r_{\mu'}(s) - P_{\mu'}^k l_{\mu'}(s)r(\mu)) \geq v_\mu(s) - P_{\mu'}^K v_\mu(s).$$

Divide both sides by $\sum_{k=0}^{K-1} P_{\mu'}^k l_{\mu'}(s)$ and take $K \to \infty$:

$$\lim_{K\to\infty} \frac{1}{\sum_{k=0}^{K-1} P_{\mu'}^k l_{\mu'}(s)} \sum_{k=0}^{K-1}(P_{\mu'}^k r_{\mu'}(s) - P_{\mu'}^k l_{\mu'}(s)r(\mu)) \geq \lim_{K\to\infty} \frac{1}{\sum_{k=0}^{K-1} P_{\mu'}^k l_{\mu'}(s)}(v_\mu(s) - P_{\mu'}^K v_\mu(s)).$$

For the l.h.s.:

$$\lim_{K\to\infty} \frac{1}{\sum_{k=0}^{K-1} P_{\mu'}^k l_{\mu'}(s)} \sum_{k=0}^{K-1}(P_{\mu'}^k r_{\mu'}(s) - P_{\mu'}^k l_{\mu'}(s)r(\mu))) = \lim_{K\to\infty} \frac{\sum_{k=0}^{K-1} P_{\mu'}^k r_{\mu'}(s)}{\sum_{k=0}^{K-1} P_{\mu'}^k l_{\mu'}(s)} - r(\mu) = r(\mu') - r(\mu).$$

For the r.h.s.:

$$\lim_{K\to\infty} \frac{1}{\sum_{k=0}^{K-1} P_{\mu'}^k l_{\mu'}(s)}(v_\mu(s) - P_{\mu'}^K v_\mu(s)) = 0.$$

Therefore $r(\mu') - r(\mu) \geq 0$.

Finally, note that a strict inequality holds if the probability of interruption when following policy $\mu'$ is non-zero. □

# B    Additional Empirical Results

## B.1    Inter-option Learning

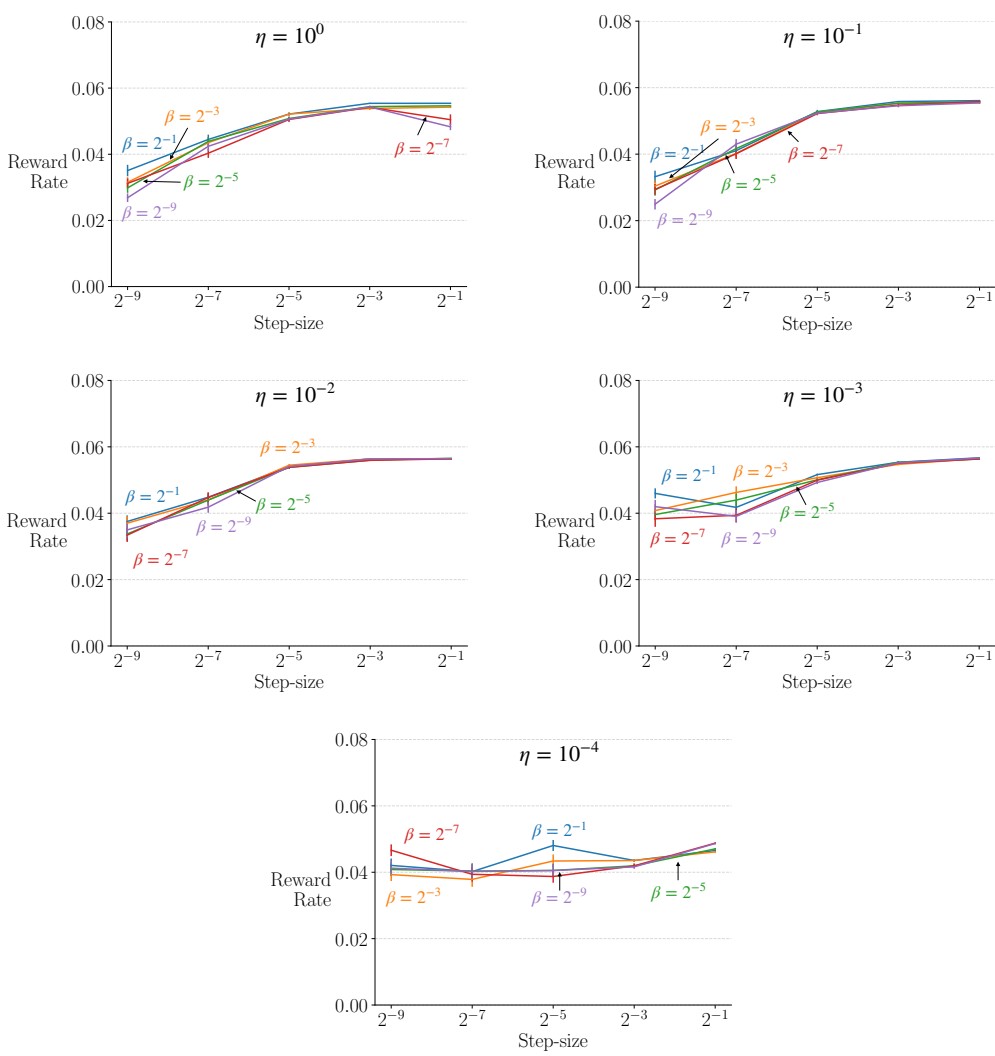

Figure B.1: Plots showing a parameter study for inter-option Differential Q-learning and the set of options $\mathcal{O} = \mathcal{H} + \mathcal{A}$ in the continuing Four-Room domain when the goal was to go to G1. Same experimental setups are used as what was described in Section 3. The x-axis denotes step size $\alpha$; the y-axis denotes the rate of the rewards averaged over all 200,000 steps of training, reflecting the rate of learning. The error bars denote one standard error. The algorithm's rate of learning varied little over a broad range of its parameters $\alpha, \beta$ and $\eta$. Small standard error bars show that the algorithm's performance varied little over multiple runs.

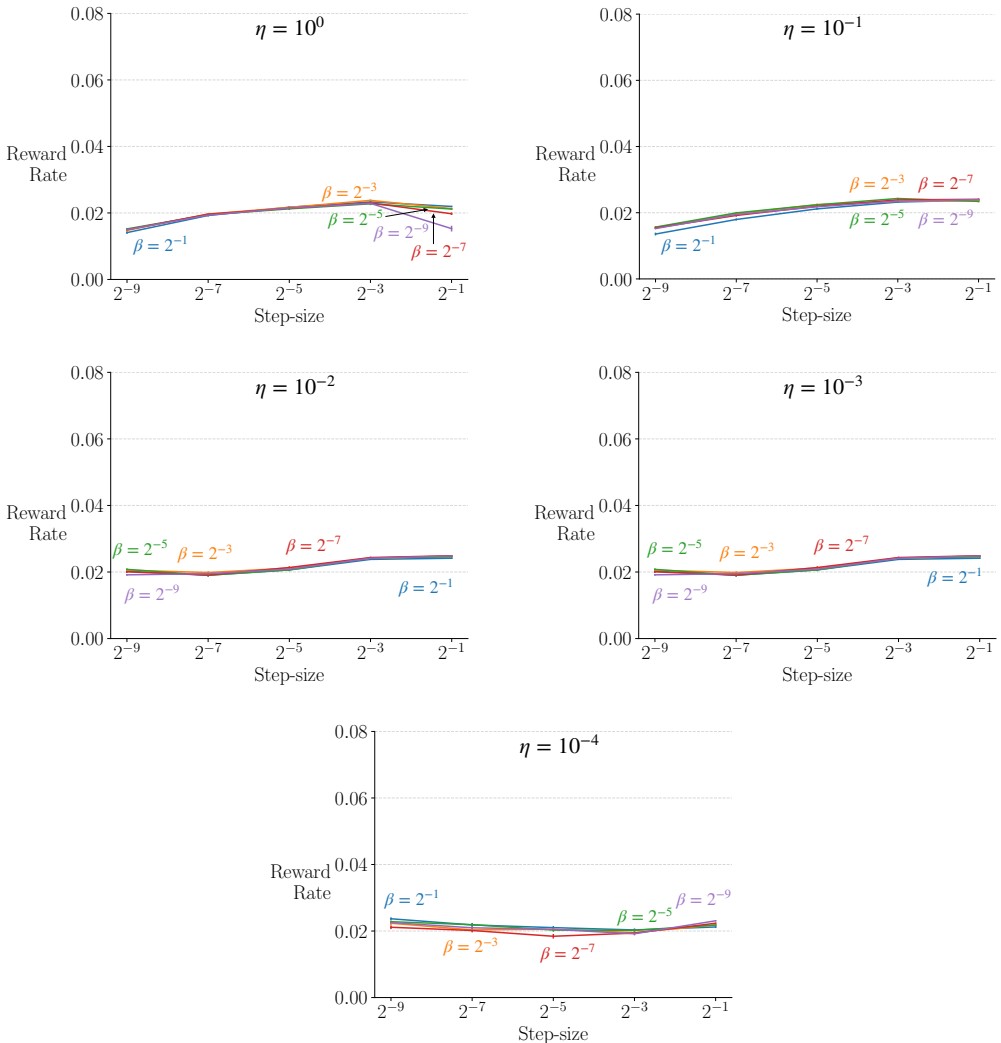

Figure B.2: Plots showing a parameter study for inter-option Differential Q-learning and the set of options $\mathcal{O} = \mathcal{H}$ in the continuing Four-Room domain when the goal was to go to G1. The experimental setting and the plot axes are the same as mentioned in Figure B.1. Compared with Figure B.1, it can be seen that the algorithm's rate of learning with $\mathcal{O} = \mathcal{H}$ was worse than it with $\mathcal{O} = \mathcal{H} + \mathcal{A}$. This is because there is no hallway option from $\mathcal{H}$ can takes the agent to G1. The algorithm's rate of learning varied little over a broad range of its parameters $\alpha, \beta$ and $\eta$, and also varied little over multiple runs.

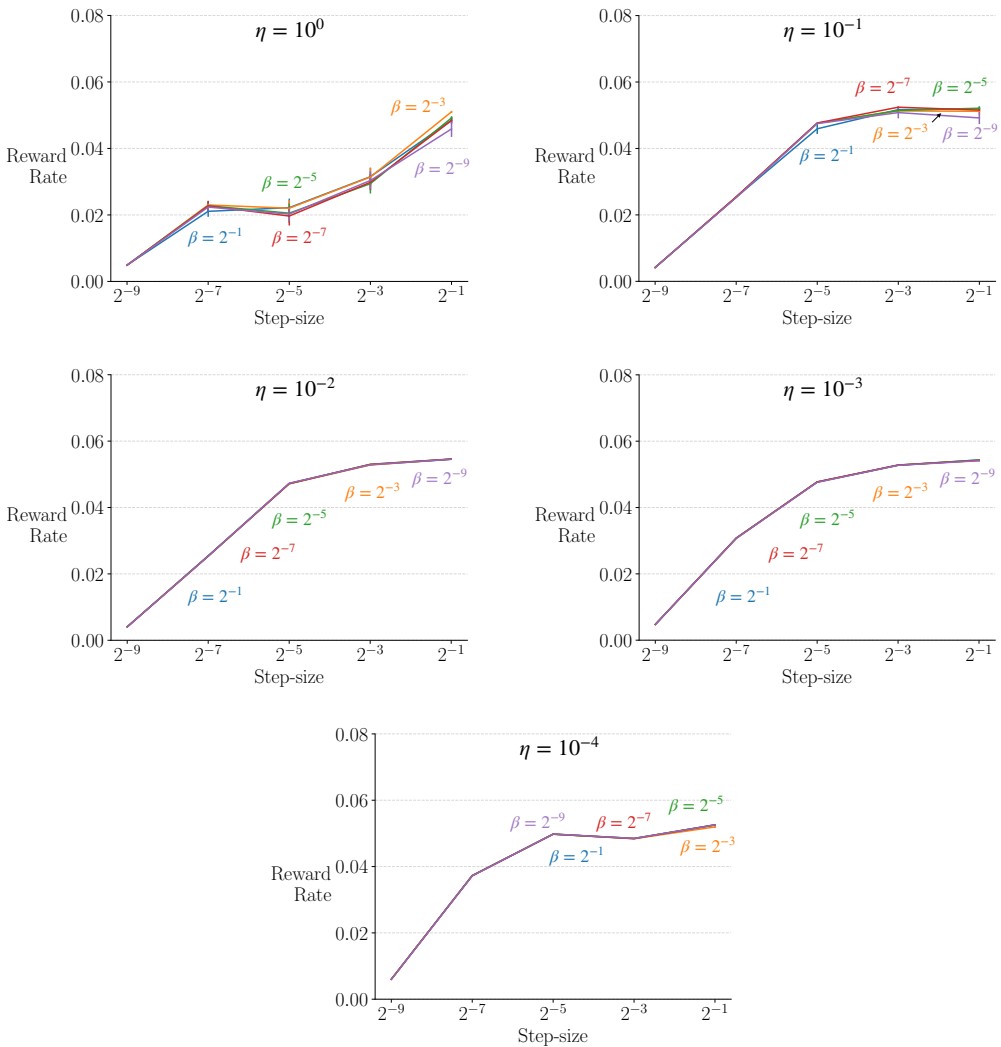

Figure B.3: Plots showing a parameter study for inter-option Differential Q-learning and the set of options $\mathcal{O} = \mathcal{A}$ in the continuing Four-Room domain when the goal was to go to $\texttt{G1}$. Note that with options being primitive actions, the algorithm becomes exactly the same as Differential Q-learning by Wan et al. (2021). The experimental setting and the plot axes are the same as mentioned in Figure B.1. Compared with Figure B.1, it can be seen that the algorithm's rate of learning with $\mathcal{O} = \mathcal{A}$ was worse than it with $\mathcal{O} = \mathcal{H} + \mathcal{A}$, particularly for small $\alpha$. The algorithm's rate of learning did not vary too much over a broad range of its parameters $\beta$ and $\eta$, and also varied little over multiple runs. The algorithm's performance is more sensitive to the choice of $\alpha$.

## B.2 Intra-option Q-learning

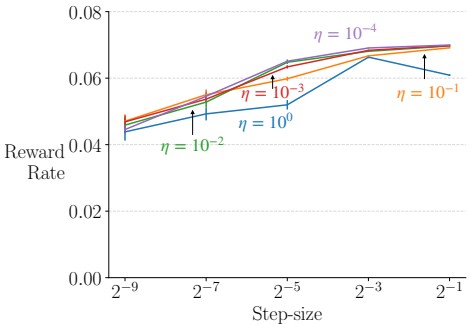

Figure B.4: Plots showing a parameter study for intra-option Differential Q-learning with the set of options $\mathcal{O} = \mathcal{H}$ in the continuing Four-Room domain when the goal was to go to `G2`. The algorithm used a behavior policy consisting only of primitive actions. The hallway options were never executed.. The experimental setting and the plot axes are the same as mentioned in Section 4. The algorithm's rate of learning varied little over a broad range of its parameters $\alpha$ and $\eta$, and also varied little over multiple runs.

## B.3 Interruption

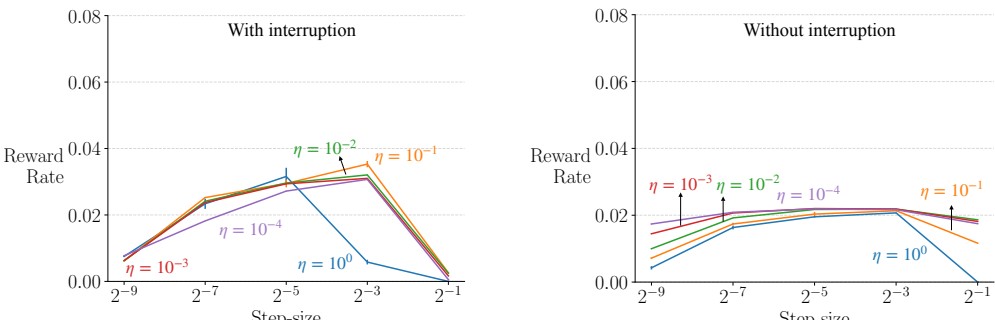

Figure B.5: Plots showing parameter studies for intra-option Differential Q-learning with and without interruption in the continuing Four-Room domain when the goal was to go to `G3`. The algorithm used the set of hallway options $\mathcal{O} = \mathcal{H}$. The experimental setting and the plot axes are the same as mentioned in Section 6. The algorithm's rate of learning with interruption was higher than it without interruption for medium sized choices of $\alpha$. When a large or small $\alpha$ was used, interruption produced a worse rate of learning. The algorithm's rate of learning varied not too much over a broad range of its parameters $\eta$ and varied little over multiple runs, regardless of interruption. The algorithm's rate of learning was more sensitive to $\alpha$ when interruption is used.

## B.4 Prediction Experiments

We also performed a set of experiments to show that both inter- and intra-option Differential Q-evaluation can learn the reward rate well. The tested environment is the same as the one used to test inter-option Differential Q-learning (with `G1`). The set of options consists of 4 primitive actions and 8 hallway options. For each state, the behavior policy randomly picks an option. The target policy is an optimal policy, which induces a reward rate 0.0625. We ran both inter- and intra option Differential Q-evaluation in this problem. The parameters used are the same with those used in inter- and intra-option Differential Q-learning experiments. The sensitivity of the two algorithms w.r.t. the parameters is shown in Figure B.6 and Figure B.7. Inter-option algorithm's reward rate error is quite robust to $\beta$. Intra-option algorithm's reward rate error is generally better than Inter-option algorithm's reward rate error unless a large stepsize $\alpha = 0.5$ is used.

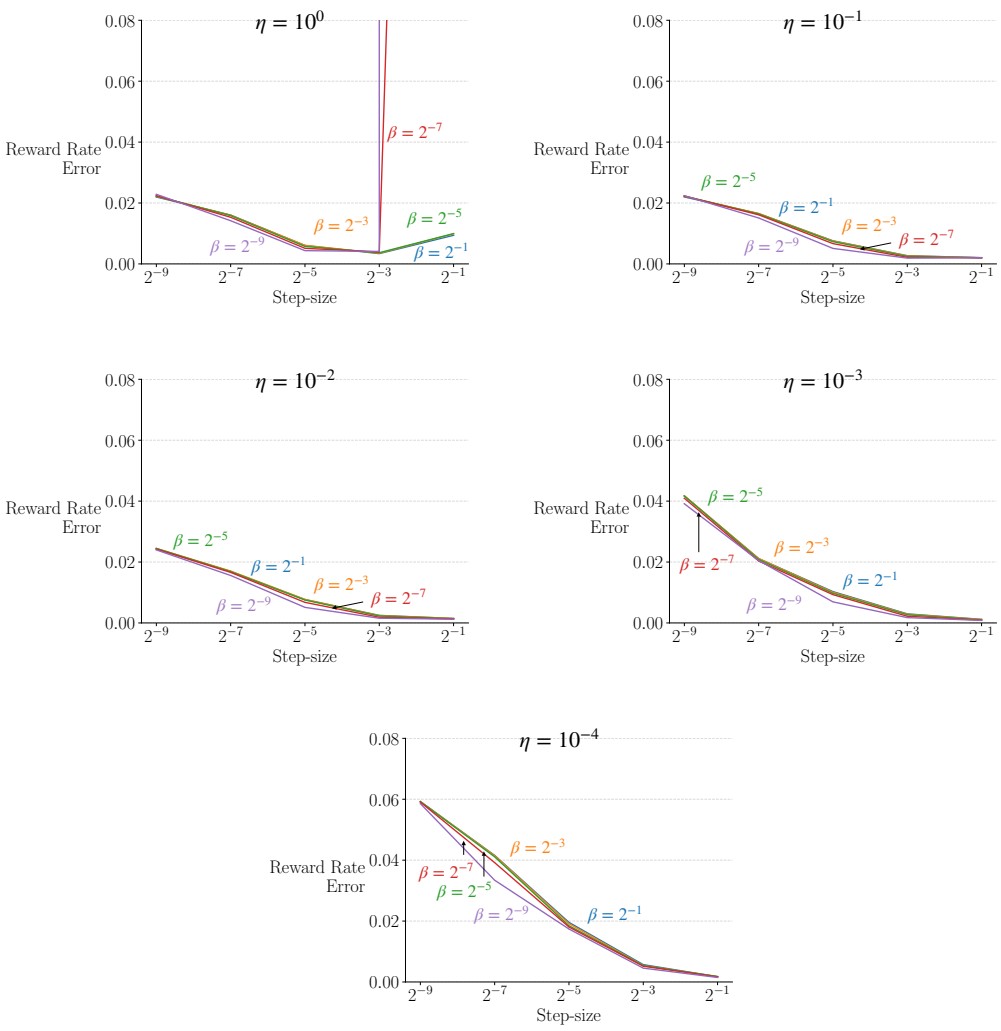

Figure B.6: Plots showing parameter studies for inter-option Differential Q-evaluation in the continuing Four-Room domain when the goal was to go to G1. The algorithm used the set of primitive actions and the set of hallway options $\mathcal{O} = \mathcal{A} + \mathcal{H}$. The y-axis is the absolute difference between the optimal reward rate 0.0625 and the estimated reward rate, averaged over all 200,000 steps.

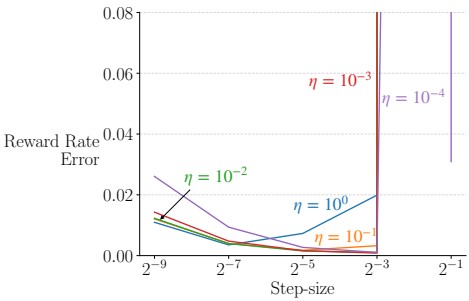

Figure B.7: Plots showing parameter studies for intra-option Differential Q-evaluation in the continuing Four-Room domain when the goal was to go to G1. The setting is the same as the one used for intra-option Differential Q-evaluation.

## C   Additional Discussion

### C.1   Two Failed Attempts on Extending Differential Q-learning to an Inter-option Algorithm

The authors have tried two other ways of extending Differential Q-learning to an Inter-option Algorithm (cf. Section 3). While these two ways appear to work properly at the first glance, they do not actually. We now show these two approaches and explain why they do not work properly.

The first extension uses, for each option, the average-reward rate per-step instead of the total reward as the reward of the option. In particular, such an extension use update rules (3) and (4), but with TD error defined as:

$$\delta_n' \doteq \hat{R}_n/\hat{L}_n - \bar{R}_n + \max_o Q_n(\hat{S}_{n+1}, o) - Q_n(\hat{S}_n, \hat{O}_n) \tag{C.1}$$

Unfortunately, such an extension can not guarantee convergence to a desired point. Specifically, the extension, if converges, will converge to a solution of $\mathbb{E}[\delta_n'] = 0$, which is not necessarily a solution of the Bellman equation $\mathbb{E}[\delta_n] = 0$ (Equation 2).

An alternative approach to avoid the instability issue is to shrink the entire update, not the option's cumulative reward, by the sample length:

$$Q_{n+1}(\hat{S}_n, \hat{O}_n) \doteq Q_n(\hat{S}_n, \hat{O}_n) + \alpha_n \delta_n/\hat{L}_n, \tag{C.2}$$

$$\bar{R}_{n+1} \doteq \bar{R}_n + \eta \alpha_n \delta_n/\hat{L}_n. \tag{C.3}$$

Still, the above two updates can not guarantee convergence to the desired values because, again, $\mathbb{E}[\delta_n/\hat{L}_n] = 0$ does not imply that the Bellman equation $\mathbb{E}[\delta_n] = 0$ is satisfied.

### C.2   Pseudocodes

---

**Algorithm 1:** Inter-option Differential Q-learning

---

**Input:** Behavioral policy $\mu_b$'s parameters (e.g., $\epsilon$ for $\epsilon$-greedy)
**Algorithm parameters:** step-size parameters $\alpha, \eta, \beta$

1 Initialize $Q(s, o) \forall s \in \mathcal{S}, o \in \mathcal{O}, \bar{R}$ arbitrarily (e.g., to zero); $L(s, o) \leftarrow 1 \forall s \in \mathcal{S}, o \in \mathcal{O}$
2 Obtain initial $S$
3 **while** *still time to train* **do**
4      Initialize $\hat{L} \leftarrow 0, \hat{R} \leftarrow 0, S_{tmp} \leftarrow S$
5      $O \leftarrow$ option sampled from $\mu_b(\cdot \mid S)$
6      **do**
7          Sample primitive action $A \sim \pi(\cdot \mid S, O)$
8          Take action $A$, observe $R, S'$
9          $\hat{L} \leftarrow \hat{L} + 1$
10         $\hat{R} \leftarrow \hat{R} + R$
11         $S \leftarrow S'$
12      **while** *$O$ doesn't terminate in $S'$*
13      $S \leftarrow S_{tmp}$
14      $L(S, O) \leftarrow L(S, O) + \beta(\hat{L} - L(S, O))$
15      $\delta \leftarrow \hat{R} - \bar{R} \cdot L(S, O) + \max_o Q(S', o) - Q(S, O)$
16      $Q(S, O) \leftarrow Q(S, O) + \alpha\delta/L(S, O)$
17      $\bar{R} \leftarrow \bar{R} + \eta\alpha\delta/L(S, O)$
18      $S \leftarrow S'$
19 **end**
20 return $Q$

---

**Algorithm 2:** Inter-option Differential Q-evaluation (learning)

---

**Input:** Behavioral policy $\mu_b$, target policy $\mu$
**Algorithm parameters:** step-size parameters $\alpha, \eta, \beta$

1  Initialize $Q(s, o) \ \forall \, s \in \mathcal{S}, o \in \mathcal{O}, \bar{R}$ arbitrarily (e.g., to zero); $L(s, o) \leftarrow 1 \ \forall \, s \in \mathcal{S}, o \in \mathcal{O}$
2  Obtain initial $S$
3  **while** *still time to train* **do**
4       Initialize $\hat{L} \leftarrow 0, \hat{R} \leftarrow 0, S_{tmp} \leftarrow S$
5       $O \leftarrow$ option sampled from $\mu_b(\cdot \mid S)$
6       **do**
7           Sample primitive action $A \sim \pi(\cdot \mid S, O)$
8           Take action $A$, observe $R, S'$
9           $\hat{L} \leftarrow \hat{L} + 1$
10          $\hat{R} \leftarrow \hat{R} + R$
11          $S \leftarrow S'$
12      **while** *$O$ doesn't terminate in $S'$*
13      $S \leftarrow S_{tmp}$
14      $L(S, O) \leftarrow L(S, O) + \beta\big(\hat{L} - L(S, O)\big)$
15      $\delta \leftarrow \hat{R} - \bar{R} \cdot L(S, O) + \sum_o \mu(o \mid S')Q(S', o) - Q(S, O)$
16      $Q(S, O) \leftarrow Q(S, O) + \alpha\delta/L(S, O)$
17      $\bar{R} \leftarrow \bar{R} + \eta\alpha\delta/L(S, O)$
18      $S \leftarrow S'$
19 **end**
20 **return** $Q$

---

**Algorithm 3:** Intra-option Differential Q-learning

---

**Input:** Behavioral policy $\mu_b$'s parameters (e.g., $\epsilon$ for $\epsilon$-greedy)
**Algorithm parameters:** step-size parameters $\alpha, \eta$

1  Initialize $Q(s, o) \ \forall \, s \in \mathcal{S}, o \in \mathcal{O}, \bar{R}$ arbitrarily (e.g., to zero)
2  Obtain initial $S$
3  **while** *still time to train* **do**
4       $O \leftarrow$ option sampled from $\mu_b(\cdot \mid S)$
5       **do**
6           Sample primitive action $A \sim \pi(\cdot \mid S, O)$
7           Take action $A$, observe $R, S'$
8           $\Delta = 0$
9           **for** *all options $o$* **do**
10              $\rho \leftarrow \pi(A \mid S, o)/\pi(A \mid S, O)$
11              $\delta \leftarrow R - \bar{R} + \Big(\big(1 - \beta(S', o)\big)Q(S', o) + \beta(S', o)\max_{o'} Q(S', o')\Big) - Q(S, o)$
12              $Q(S, o) \leftarrow Q(S, o) + \alpha\rho\delta$
13              $\Delta \leftarrow \Delta + \eta\alpha\rho\delta$
14          **end**
15          $\bar{R} \leftarrow \bar{R} + \Delta$
16          $S \leftarrow S'$
17      **while** *$O$ doesn't terminate in $S$*
18 **end**
19 **return** $Q$

**Algorithm 4:** Intra-option Differential Q-learning with interruption

**Input:** Behavioral policy $\mu_b$'s parameters (e.g., $\epsilon$ for $\epsilon$-greedy)
**Algorithm parameters:** step-size parameters $\alpha, \eta$

1   Initialize $Q(s, o) \ \forall \, s \in \mathcal{S}, o \in \mathcal{O}, \bar{R}$ arbitrarily (e.g., to zero)
2   Obtain initial $S$
3   $O \leftarrow$ option sampled from $\mu_b(\cdot|S)$
4   **while** *still time to train* **do**
5      **if** $O \notin \operatorname{argmax} Q(S, \cdot)$ **then**
6         $O \leftarrow$ option sampled from $\mu_b(\cdot|S)$
7      **end**
8      Sample primitive action $A \sim \pi(\cdot|S, O)$
9      Take action $A$, observe $R, S'$
10     $\Delta = 0$
11     **for** *all options o* **do**
12       $\rho \leftarrow \pi(A|S, o)/\pi(A|S, O)$
13       $\delta \leftarrow R - \bar{R} + \Big( \big(1 - \beta(S', o)\big)Q(S', o) + \beta(S', o)\max_{o'} Q(S', o') \Big) - Q(S, o)$
14       $Q(S, o) \leftarrow Q(S, o) + \alpha\rho\delta$
15       $\Delta \leftarrow \Delta + \eta\alpha\rho\delta$
16     **end**
17     $\bar{R} \leftarrow \bar{R} + \Delta$
18     $S = S'$
19   **end**
20   return $Q$

---

**Algorithm 5:** Intra-option Differential Q-evaluation (learning)

**Input:** Behavioral policy $\mu_b$, target policy $\mu$
**Algorithm parameters:** step-size parameters $\alpha, \eta$

1   Initialize $Q(s, o) \ \forall \, s \in \mathcal{S}, o \in \mathcal{O}, \bar{R}$ arbitrarily (e.g., to zero)
2   Obtain initial $S$
3   **while** *still time to train* **do**
4      $O \leftarrow$ option sampled from $\mu_b(\cdot \mid S)$
5      **do**
6        Sample primitive action $A \sim \pi(\cdot \mid S, O)$
7        Take action $A$, observe $R, S'$
8        $\Delta = 0$
9        **for** *all options o* **do**
10         $\rho \leftarrow \pi(A \mid S, o)/\pi(A \mid S, O)$
11         $\delta \leftarrow R - \bar{R} + \Big( \big(1 - \beta(S', o)\big)Q(S', o) + \beta(S', o)\sum_{o'} \mu(o' \mid S')Q(S', o') \Big) - Q(S, o)$
12         $Q(S, o) \leftarrow Q(S, o) + \alpha\rho\delta$
13         $\Delta \leftarrow \Delta + \eta\alpha\rho\delta$
14        **end**
15        $\bar{R} \leftarrow \bar{R} + \Delta$
16        $S \leftarrow S'$
17      **while** *O doesn't terminate in S*
18   **end**
19   return $Q$

**Algorithm 6:** Combined Algorithm: Intra-option Model-learning + Inter-option Q-planning

---

**Input:** Behavioral policy $\mu_b$'s parameters (e.g., $\epsilon$ for $\epsilon$-greedy)

**Algorithm parameters:** step-size parameters $\alpha, \beta, \eta$; number of planning steps per time step $n$

**1** Initialize $Q(s, o), P(x \mid s, o), R(s, o) \; \forall \, s, x \in \mathcal{S}, o \in \mathcal{O}, \bar{R}$, arbitrarily (e.g., to zero);
   $L(s, o) = 1 \; \forall \, s \in \mathcal{S}, o \in \mathcal{O}; T \leftarrow$ False

**2** **while** *still time to train* **do**

**3** $\quad$ $S \leftarrow$ current state

**4** $\quad$ $O \leftarrow$ option sampled from $\mu_b(\cdot \mid S)$

**5** $\quad$ **while** *$T$ is False* **do**

**6** $\quad\quad$ Sample primitive action $A \sim \pi(\cdot \mid S, O)$

**7** $\quad\quad$ Take action $A$, observe $R', S'$

**8** $\quad\quad$ **for** *all options $o$ such that $\pi(A \mid S, o) > 0$* **do**

**9** $\quad\quad\quad$ $\rho \leftarrow \pi(A \mid S, o) / \pi(A \mid S, O)$

**10** $\quad\quad\quad$ **for** *all states $x \in \mathcal{S}$* **do**

**11** $\quad\quad\quad\quad$ $P(x \mid S, o) \leftarrow P(x \mid S, o) + \beta\rho\Big(\beta(S', o)\mathbb{I}(S' = x) + \big(1 - \beta(S', o)\big)P(x \mid$
$\quad\quad\quad\quad\quad S', o) - P(x \mid S, o)\Big)$

**12** $\quad\quad\quad$ **end**

**13** $\quad\quad\quad$ $R(S, o) \leftarrow R(S, o) + \beta\rho\Big(R' + \big(1 - \beta(S', o)\big)R(S', o) - R(S, o)\Big)$

**14** $\quad\quad\quad$ $L(S, o) \leftarrow L(S, o) + \beta\rho\Big(1 + \big(1 - \beta(S', o)\big)L(S', o) - L(S, o)\Big)$

**15** $\quad\quad$ **end**

**16** $\quad\quad$ $T \leftarrow$ indicator of termination sampled from $\beta(S', O)$

**17** $\quad\quad$ **for** *all of the $n$ planning steps* **do**

**18** $\quad\quad\quad$ $S \leftarrow$ a random previously observed state

**19** $\quad\quad\quad$ $O \leftarrow$ a random option previously taken in $S$

**20** $\quad\quad\quad$ $S' \leftarrow$ a sampled state from $P(\cdot \mid S, O)$

**21** $\quad\quad\quad$ $\delta \leftarrow R(S, O) - L(S, O)\bar{R} + \max_o Q(S', o) - Q(S, O)$

**22** $\quad\quad\quad$ $Q(S, O) \leftarrow Q(S, O) + \alpha\rho\delta/L(S, O)$

**23** $\quad\quad\quad$ $\bar{R} \leftarrow \bar{R} + \eta\alpha\rho\delta/L(S, O)$

**24** $\quad\quad$ **end**

**25** $\quad$ **end**

**26** **end**

**27** return $Q$

---