# OpenReview forum: "Average-Reward Learning and Planning with Options"
_NeurIPS.cc/2021/Conference — NeurIPS 2021 Poster_

### Official Review · Reviewer_Dp3F · 2021-07-11

**Rating:** 6
**Confidence:** 4

**Summary:**

This paper extended the learning and planning algorithms and their theoretical properties for options framework in discounted settings MDP settings to average-reward MDPs. The main contributions include three algorithms their convergence property. These algorithms include off-policy inter-option learning and planning, intra-option value learning and planning, intra-option model learning and planning. The convergence property is justified based on the resulted developed for differential Q-learning, an off-policy control algorithm. The paper also extended the notion of option-interrupting behavior from the discounted case to the average-reward formulation and showed it’s efficacy on a continuing version of the 4room domain. Importantly, the intra-option differential Q-learning introduced in this paper can be sued with options having stochastic policies (by using the importance sampling ratio), whereas the original discounted intra-option learning methods were restricted to options with deterministic policies.

**Ethical Concerns:**

I don't have any ethical concerns.

**Limitations And Societal Impact:**

The limitations of this work has been discussed in the last section of the paper.

**Main Review:**

In all the paper is well written and provides some didactic results for options learning and planning in average-reward MDP settings which are complementary to the existing literature on the same topic but mostly for discounted MDP settings.

One issue I have is that many theoretical results seem are directly and trivially extend from some existing work, e.g. Wan et al.’s differential Q learning and Wan et al.’s differential learning and planning algorithms for average-reward MDPs. The paper would be stronger if the difficulties of the theoretical results that are unique to the average-reward settings for options can be highlighted. Another issue is the sufficiency of empirical study. Admittedly the paper is mostly for didactic purpose and the algorithms are developed based on tabular representations. It would be better if more baselines can be compared. i.e,  in figure 4, maybe the authors can include some method from discounted case, but set the discount factor to be large.  Also there is only one domain used for empirical study, it might be stronger if results based on a problem different from 4rooms and described by average reward MDP can be provided.


**Time Spent Reviewing:**

3h

---

> ### Author Response · Authors · 2021-08-11
> **Response to Reviewer Dp3F**
>
> Thank you for taking the time to review the paper. We are glad you felt the paper was well-written. We also want to thank you for highlighting a possibly under-valued contribution of going beyond the original options framework to handle stochastic option policies. We have responded to your comments one-by-one:
>
> **Comment:**  “The paper would be stronger if the difficulties of the theoretical results that are unique to the average-reward settings for options can be highlighted.”
>
> **Response:** We agree that the technical difficulties, as well as the originality and importance of our contributions, are not explicitly highlighted in the current manuscript. We will add more discussion about it in the revision. For instance:
> We generalize Wan et al.’s proof techniques to show convergence for both the cases of actions and options, as well as inter-and intra-option algorithms. We feel it is quite an elegant contribution to have such a unified proof for all the cases.
> The options case requires special treatment on the algorithms side. For example, in the case of primitive action, there is no need to learn the option length; there are no inter-and intra-option algorithms.
>
> **Comment:** “It would be better if more baselines can be compared. i.e, in figure 4, maybe the authors can include some method from the discounted case, but set the discount factor to be large.”
>
> **Response:** Our intent is to extend options to the average-reward setting, not compare the average-reward setting to the discounted one (in retrospect, we feel we did not make this sufficiently clear, and we plan to make it more so in the revision). From this point of view, a comparison with a solution designed for the discounted setting would be inappropriate and difficult to interpret.
>
> **Comment:** “Also there is only one domain used for empirical study, it might be stronger if results based on a problem different from 4rooms and described by average reward MDP can be provided.”
>
> **Response:**  The experiments in the paper currently serve to validate the theoretical contributions. We agree that further empirical demonstrations on domains of varying complexities might improve the confidence in the proposed algorithms’ efficacy — this is one of the most natural directions of future work.
>
> We thank you again for your time. Do let us know if you have any follow-up questions or concerns.
>
> **References**
>
> Wan, Y., Naik, A., & Sutton, R. S. (2021). Learning and Planning in Average-Reward Markov Decision Processes

---

> > ### Comment · Reviewer_Dp3F · 2021-08-30
> > **My review of the paper remains same**
> >
> > I have read the authors' rebuttal and appreciate their efforts addressing my comments. My review and rating of the paper remains the same.

---

### Official Review · Reviewer_TT1t · 2021-07-13

**Rating:** 8
**Confidence:** 4

**Summary:**

This paper extends the Options framework from the discounted MDP setting to the average-reward setting. The authors introduce 4 main contributions: (1) an off-policy inter-option learning algorithm with convergence guarantees; (2) intra-option techniques for learning value functions and models; (3) a method that allows for these ideas to be deployed under sample-based planning algorithms; and (4) an extended definition of interrupting options, adapting its original definition (as proposed for the discounted case) to the average-reward formulation.


**Limitations And Societal Impact:**

My understanding is that this paper's main contributions are theoretical. The paper does show a few simple experiments, but additional experimental analysis (even for these simple domains) could be made; see, for instance, my comment #9, above. In terms of technical limitations, I believe that the main one is that the paper tackles only the tabular setting, which is usually not of particular interest for real-world applications. The authors, however, do acknowledge this, and propose extending these ideas to the case of linear function approximation as future work.

**Main Review:**

This is a well-written paper that makes an important set of contributions to the area. Average-reward RL has been (comparatively) less studied than the discounted setting, but it is equally important. Being able to extend important key concepts in Hierarchical RL to the average reward formulation is certainly relevant to the community.

Besides being significant, I found the paper to be clear and technically sound. I have a few questions and suggestions:

1) when defining the reward rate of a policy, r^C(mu), what does the "C" superscript stand for? I do  understand the difference between r^C(mu) and r(mu), but was confused about the terminology.

2) regarding Assumption 1: why are all stationary policies in the proposed MDP recurrent? Is this a well-known result? If so, a citation would be helpful. If not: is this statement a claim (that can be proved) or part of the assumption itself?

3) in Equations 1 and 2, \bar{r} was never defined, I believe.

4) in Equation 2, shouldn't it be q*(s,o) instead of q(s,o)?

5) regarding the technique presented in Section 3: my understanding is that the main contribution, here, is the idea of scaling the step-size not by the observed option length, but by an estimate (model) of the option's length. Is that the case? If so, what is the theoretical justification for doing this? Is this meaningful because the option length model \hat{L} encodes the expected length, and using that (the expectation, instead of individual length realizations) decreases the variance of the update rule? A discussion motivating this idea would be helpful.

6) even though the discussion of Equations 6-9 is clear, it was not immediately clear to me how all of these ideas could be combined to construct a planning algorithm. I suggest including high-level pseudo-code, or at least a discussion (in English) about the overall steps involved.

7) regarding the results presented in Figure 2: is the setting of O=A+H faster than the alternatives because options allowed the agent to explore more efficiently, during the early stages of its life? If so, were the options helpful *only* during the initial exploratory phase? One simple way of checking whether options were helpful mostly with exploration, or if they were used by the agent to express optimal policies, is to check whether the final learned policy made use of options or, alternatively, whether it used mostly primitives. A discussion about this would be helpful.

8) still regarding Figure 2, the authors claim the the algorithm's rate of learning varied little over a broad range of parameters. However, it seems like the reward rate can vary from approximately 0.03 up to 0.06. This is twice the reward rate, which is not a little change. I suggest either revisiting this claim or better supporting it.

9) Figure 3 shows the parameter-sensitivity of Gosavi's algorithm. Why did you not include an analysis of its reward rate, like you did in Fig2a when analyzing your proposed algorithm?

10) in Equation 17, where is s' coming from? Perhaps the second summation should be sum_{s', r'} instead of sum_r?

11) in Theorem 6, what is v_mu(s)? This was never defined, I believe


**Time Spent Reviewing:**

4

---

> ### Author Response · Authors · 2021-08-11
> **Response to Reviewer TT1t**
>
> Thank you for taking the time to review the paper and recommending a clear accept. We are glad that you concur with the importance of extending the options framework to the under-studied but important average-reward formulation. We shall incorporate your helpful suggestions for making the paper even better:
>
> 1. The C denotes “continuing” and originates from Puterman’s MDP textbook (1994: Eq 11.4.2).
> 2. All stationary policies in the proposed MDP being recurrent is part of the assumption. It ensures that all states occur an infinite number of times in a single stream of the learning agent’s experience. For states that only occur a finite number of times (transient states), no learning algorithm can be guaranteed to estimate their values correctly from a single stream of experience.
> 3. In (1) and (2), $\bar{r}$ and $q$ are variables whose solutions are described in the following paragraph (line 98 onwards) in terms of quantities defined earlier.
> 4. In (2), $q$ indeed denotes a solution to the Bellman optimality equation for the average-reward case. In the discounted case, the solution is unique and is usually denoted by $q^*$. In the average-reward case, though, there are infinitely many solutions to the Bellman optimality (as well as evaluation) equations (all offset by a constant). Hence we used a generic symbol $q$. We shall point this out in the paper and perhaps add a citation to Puterman’s (1994) textbook for more details.
> 5. Yes, the scaling factor is the estimated option length. The reason is that the expected length produces an appropriate update while the sample length does not.  We discuss this briefly in the paragraph starting on line 115. Crucially, the fixed points of the update scaled by the sample length, if they exist, are not necessarily solutions to the Bellman equation (eq 2). On the other hand, scaling the update by the expected length results in an update whose fixed points are solutions to the Bellman equation. More details are provided in Appendix C.1.
> 6. Due to the main paper’s page limit, we had to add the pseudocodes for all the proposed algorithms in Appendix C. We would be happy to discuss any further questions about the pseudocodes.
> 7. This is a good point! In this experiment, the optimal reward rate with $\mathcal O= \mathcal A$ and with $\mathcal O=\mathcal A+ \mathcal H$ is the same. Learned policies picking either primitive actions or options are optimal. In general, optimal policies might consist of helpful options. It would be interesting to check the statistics of option usage for exploration and exploitation over the training period.
> 8. Another good point: Figure 2’s caption should just state robustness to eta. The claim of relative robustness to alpha can be towards the end of the section when compared to the baseline’s larger range of around 0.01 to 0.06. Thank you for pointing this out, we will make this change in the revision.
> 9. With Figure 2 (left), we wanted to demonstrate that the empirical observations using different sets of options in the average-reward setting echo the corresponding observations in the discounted setting (Sutton et al. 1999). The sensitivity plots for our inter-option algorithm and Gosavi’s algorithm are shown in Figure 2 (right) and Figure 3, which characterize the overall average reward obtained by the agent over the training period.
> 10. Indeed, the second summation should be over ${s’,r}$. Thank you for pointing it out!
> 11. $v_\mu(s)$ is the value of state $s$ under policy $\mu$: $v_{\mu}(s) \doteq \sum_o q_\mu(s, o)$. We missed defining it in the paper; thanks for pointing it out!
>
> We thank you again for the helpful suggestions for improving the paper. Please let us know if you have any follow-up questions or feedback.
>
> **References**
>
> Gosavi, A. (2004). Reinforcement Learning for Long-run Average Cost.
>
> Puterman, M. L. (1994). Markov Decision Processes: Discrete Stochastic Dynamic Programming.
>
> Sutton, R. S., Precup, D., & Singh, S. (1999). Between MDPs and Semi-MDPs: A Framework for Temporal Abstraction in Reinforcement Learning.

---

> > ### Comment · Reviewer_TT1t · 2021-08-31
> > **Comments on authors' rebuttal**
> >
> > I would like to thank the authors for carefully responding to my questions, concerns, and criticisms. I believe they have addressed all my main concerns. I also appreciate their willingness to update the presentation of the paper to try to make their contributions clearer to the overall community. My review and rating of the paper remain the same and maintain my position that this is a well-written paper that makes an important contribution to the area.

---

### Official Review · Reviewer_ufJp · 2021-07-16

**Rating:** 6
**Confidence:** 3

**Summary:**

This paper extends options framework to average-reward Markov Decision Processes. This paper presents inter-option learning and planning algorithms, and intra-option learning and planning algorithms, as well as provide theoretical analyses that support these algorithms. It also uses an idea of allowing interruption during the option, which allows behaviors with a potentially higher than without interruption - and combine it in its framework.

**Main Review:**

Detailed summary
- The introduction part explains the concept of RL, options framework, SMDP, inter-/intra-option algorithms, etc. I think the intro part was well written, but it would've been a bit better if it contained a brief background explanation about average-reward MDP either in section 1 or in section 2.
- Inter-option: Q_{n+1}(S_n, O_n) = Q_n(S_n, O_n) + \alpha_n \delta_n / L_n(S_n, O_n), where L approximates the expected length of s-o pairs. L_n is also updated by using another step-size \beta_n. The L_n(s_n, o_n) is different from the sampled option length.
- I've checked the convergence proofs for inter-options, and didn't find any flaws
- Intra-option: q(s,o) = \sum_{a} \pi(a|s,o) \sum_{s',r} p(s',r|s,a) (r-R + u^q(s',o)). The intra-option differential q-learning and q-evaluation algorithms are suggested. Again, the proof for theorem3 checked.
- intra-option model learning/planning: should predict the duration of execution of the option.
- Interruption improves the performance of intra-option models.

I think this paper is a well-written paper with a combinations of novel, original ideas (average-reward MDP and option framework) and mathematical approach. Although the level of experiments are not extensive, the main idea in this paper has a potential to be extended to more general cases including function approximation.

I have two questions.

Q1. General question: I get the idea of combining average-reward MDP and option framework; but other than achieving high reward empirically, is there any motivation or necessity to use average-reward MDP in the setting of option framework?

Q2. Empirical results in Appendix: the algorithm's sensitivity to hyperparameter \alpha, \beta, \eta varies; in the case of inter-option differential q-learning, the algorithm is not very sensitive to any parameter \alpha, \beta, \eta. In the case of inter-option learning with options O=A, the algorithm was more sensitive to the choice of \alpha; in the case of intra-option algorithms; the choice of \alpha and \eta didn't make much difference. Can you give us an intuition of why this difference happens in different cases?

**Time Spent Reviewing:**

7

---

> ### Author Response · Authors · 2021-08-11
> **Response to Reviewer ufJp**
>
> Thank you for taking the time to read and review the paper. We are glad that you felt the ideas were original and the writing clear. We also agree that the main ideas are general enough to be extended to the case of function approximation.
>
> **Comment:** “... is there any motivation or necessity to use average-reward MDP in the setting of option framework”:
>
> **Response:** There is a strong motivation throughout reinforcement learning for moving from the discounted objective to the average reward objective, primarily because, once genuine function approximation is employed, the discounted problem is no longer well defined and is deprecated (see Sutton & Barto 2018, Section 10.4; Naik et al. 2019). If we switch to the average-reward formulation for the basic problem, and we want the advantages of options in acting, learning, and planning, then we must extend options to the average-reward setting as well. One must be done along with the other. We realize in retrospect that we did not clearly present this motivation in the paper. Now that we see that it is missing, we will add it to the revision, early in the introduction.
>
> **Comment:** “Empirical results in Appendix ... Can you give us an intuition of why this difference happens in different cases?”:
>
> **Response:** We believe that the algorithm is more sensitive to alpha using O=A because the algorithm does not use hallways options and therefore generally learns slower. In such a case, the choice of the step size would be more significant. When the algorithm has the choice of hallways options, it learns faster and therefore a relatively smaller step size still results in a good performance.
>
> We thank you again for your time. Please let us know if you have any follow-up questions or comments.
>
> **References:**
>
> Naik, A., Shariff, R., Yasui, N., Yao, H., & Sutton, R. S. (2019). Discounted reinforcement learning is not an optimization problem.
>
> Sutton, R. S., & Barto, A. G. (2018). Reinforcement Learning: An Introduction.

---

### Official Review · Reviewer_Pn3J · 2021-07-18

**Rating:** 6
**Confidence:** 4

**Summary:**

The paper presents an extension of the options framework to the average-reward formulation in RL. To this end, the key contributions of this paper are: 1) it extends an off-policy control algorithm by Wan, Naik, Sutton’s differential q learning to inter-option differential q learning, 2) presents an incremental sample-based inter-option differential q learning planning algorithm. For 1) they guarantee convergence without requiring a special state, and prove convergence for 2) as well. 3) Next, the work presents intra-option learning and planning algorithms for the average reward formulation including their convergence guarantees. 4) Finally, this work also extends the interruption algorithm from Sutton 1999 to the average reward case. The empirical results support the theoretical claims but could be extended further.

**Ethical Concerns:**

None.

**Limitations And Societal Impact:**

Yes, limitations of the work have been discussed in Sec 7.
I do not see a broader impact statement, I encourage you to write one and add that to the paper.


**Main Review:**

**Originality and Significance**:
The work builds upon the foundations laid in Wan, Naik, Sutton (2021) such as Assumption 1, concept of reward rate and the differential value function, etc. Despite that, there are a lot of interesting and important challenges that the work addresses by extending the options framework to the average reward formulation. For instance, the instability incurred due to the large option length and consequently the dependence of the step size on the option length is a key challenge.

The ideas presented in the paper are presented with clarity and technically sound. I appreciate the inclusion of extended discussion in the appendix on the attempts for this which did not work. This gives further insight on the merits of the work.

Re option models: Can you comment on the comparison to the option multi-time-models both theoretically and empirically? Any insights would be helpful. While the duration component is an explicit additional component as presented here, the MTM captures the full distribution of all possible lengths of the options. What do we gain by explicitly modelling the length of the options? Is the key impact in the scaling component in the TD error (Alg 6, L21-23)? It would be good to expand on this impact in the main paper.


**Empirical Analysis**:  The empirical analysis provided more confidence and understanding of the ideas presented in the work and its impact on the robustness for example. The inter-option differential q learning in Fig 2 and 3 helps in establishing the improvements in robustness of the algorithm proposed towards the hyper-parameters including a comparison to the baseline. However, I have the following concerns and questions:

* The empirical analysis as it stands seems rather weak. For instance, in Fig 1 we conclude that the results indeed match theorem 1, but potentially a lot more analysis can be done to show the utility of the proposed algorithms.
* Why do we not see the Gosavi’s (2004) algorithm compared in Fig 1 left?
* I wonder if it would be valuable to compare with the primitive actions differential q learning (Wan, Naik and Sutton, 2021) as a baseline. Especially since the work derives a lot from the aforementioned paper? This could potentially highlight the trade-offs between actions and options in the average reward case.
* As also pointed out by the authors, it would be helpful to provide a stronger empirical analysis in other domains with closely related baselines. For instance, an interesting comparison can be made between the original options framework and the one present in this paper to understand and analyse the trade-off. This analysis might highlight the
* Additionally, would a baseline such as the average cost TD learning be a suitable candidate to compare with? If not, could you please shed some light on why?


**Writing and Presentation**:
I suggest expanding the introduction specifically to address the motivation for average-reward setting. It is clear that the existing literature has the discounted formulation already, but it is hard for the reader to gauge why the average-setting is important. This is easy to address by rewriting parts of the introduction and the paper.

**Miscellaneous**:
 * I was a bit confused in the figures because only the proposed algorithm is shown with little to no comparisons. The sensitivity analysis could be placed side by side perhaps to show the robustness difference.
* It took a while to understand that Fig 3 is the baseline method. Perhaps you might want to make it more explicit.


**Time Spent Reviewing:**

4

---

> ### Author Response · Authors · 2021-08-11
> **Response to Reviewer Pn3J**
>
> Thank you for taking the time to review the paper and to provide suggestions for improving its quality. Indeed, there were several challenges in extending the options framework to the understudied but important average-reward formulation. We are glad to hear you found utility in our discussion (in the appendix) of unsuccessful attempts at addressing the aforementioned challenges.
>
> **Comment:** “Can you comment on the comparison to the option multi-time-models both theoretically and empirically?”
>
> **Response:** The first challenge of using a multi-time-model (MTM) is its representation. Note that the set of possible sample lengths of an option, starting from a given state, can be infinite, even if the expected length is finite. Representing an arbitrary probability distribution of such a set of events using a finite look-up table can be challenging as well as potentially require domain-specific assumptions on the distribution.
> Even if we can represent and learn an MTM, we couldn't find an appropriate way to scale the TD-error. In particular, using a sampled option length from an MTM does not guarantee convergence to a solution of the Bellman equations (this was one of the failed attempts that we discussed in Appendix C.1).
>
> **Comment:** ”... in Fig 1 we conclude that the results indeed match theorem 1, but potentially a lot more analysis can be done to show the utility of the proposed algorithms”
>
> **Response:** The main purpose of the experiments is to validate the theoretical contributions of the paper. We agree that further empirical demonstrations on domains of varying complexities might improve the confidence in the proposed algorithms’ efficacy — this is one of the most natural directions of future work.
>
> **Comment:** “Why do we not see the Gosavi’s (2004) algorithm compared in Fig 1 left?”
>
> **Response:** With Figure 2 (left), we wanted to demonstrate that the empirical observations using different sets of options in the average-reward setting echo the corresponding observations in the discounted setting (Sutton et al. 1999). The sensitivity plots for our inter-option algorithm and Gosavi’s algorithm are shown in Figure 2 (right) and Figure 3, which characterize the overall average reward obtained by the agent over the training period.
>
> **Comment:** “I wonder if it would be valuable to compare with the primitive actions differential q learning (Wan, Naik and Sutton, 2021) as a baseline … This could potentially highlight the trade-offs between actions and options in the average reward case.”
>
> **Response:** We’ve already done that. The experiments in Section 3 use: (1) just primitive actions, (2) just options, and (3) primitive actions + options. In case (1), the updates simplify to Wan et al. (2021)’s Differential Q-learning. Correspondingly, Figure 2 (left) shows that learning with both options and primitive actions is relatively faster on this problem and parameter configuration. The given set of options here are actually helpful for the goal; we would expect the quality of the given options to play a big role in the trade-off you mentioned.
>
> **Comment:** “As also pointed out by the authors, it would be helpful to provide a stronger empirical analysis in other domains with closely related baselines. For instance, an interesting comparison can be made between the original options framework and the one present in this paper to understand and analyze the trade-off.”
>
> **Response:** Our intent is to extend options to the average-reward setting, not compare the average-reward setting to the discounted one (in retrospect, we feel we did not make this sufficiently clear, and we plan to make it more so in the revision). From this point of view, a comparison with a solution designed for the discounted setting would be inappropriate and difficult to interpret. But we agree with the larger point regarding the utility of an extensive empirical study that goes beyond validating the theory. As mentioned earlier, this is indeed a direction of future work.
>
> **Comment:** Additionally, would a baseline such as the average cost TD learning be a suitable candidate to compare with? If not, could you please shed some light on why?
>
> **Response:** Indeed, the inter-option version of Average Cost TD is a natural baseline. The reason we didn’t include a comparison with it is that the experiments in this paper focus on the control setting while Average Cost TD is a prediction algorithm. (A quick observation is that an inter-option version of Average Cost TD would be similar to the prediction version of Gosavi’s algorithm because both use the sample average of observed rewards to estimate the reward rate.) We will add prediction experiments in the appendix of the revision; they would certainly add additional value even if they are not central to the contributions of the paper.
>
> **Comment:** I suggest expanding the introduction specifically to address the motivation for average-reward setting.
>
> **Response:** Good suggestion. As suggested by other reviewers as well, we shall add a brief discussion in the introduction motivating the average-reward formulation as an alternative to the discounted formulation for continuing problems. We shall also add pointers to the literature that primarily deal with this topic (e.g., Naik et al. 2019, Sutton & Barto 2018: Ch.10, Schwartz 1993)
>
> **Comment:** “It took a while to understand that Fig 3 is the baseline method. Perhaps you might want to make it more explicit.”
>
> **Response:** We will explicitly state in Figure 3’s caption that Gosavi’s algorithm is the baseline. The main text alongside the figure already says this; together with the modified caption, the chances of any future confusion should be reduced.
>
> Hope this helps. Do let us know if you have any follow-up questions or suggestions. We thank you once again for your time!
>
> **References:**
>
> Gosavi, A. (2004). Reinforcement Learning for Long-run Average Cost.
>
> Naik, A., Shariff, R., Yasui, N., Yao, H., & Sutton, R. S. (2019). Discounted reinforcement learning is not an optimization problem.
>
> Schwartz, A. (1993). A reinforcement learning method for maximizing undiscounted rewards.
>
> Sutton, R. S., Precup, D., & Singh, S. (1999). Between MDPs and Semi-MDPs: A Framework for Temporal Abstraction in Reinforcement Learning.
>
> Sutton, R. S., & Barto, A. G. (2018). Reinforcement Learning: An Introduction.
>
> Wan, Y., Naik, A., & Sutton, R. S. (2021). Learning and Planning in Average-Reward Markov Decision Processes

---

> > ### Comment · Reviewer_Pn3J · 2021-08-20
> > **Thanks.**
> >
> > Thank you for your response to my concerns and comments. I acknowledge the rebuttal response and recommend acceptance.

---

### Decision · Program_Chairs · 2021-09-28

**Decision:**

Accept (Poster)

**Comment:**

The paper advances the state of the art for average reward hierarchical RL.  The reviewers unanimously recommended acceptance.  The work is theoretically solid and novel.  Well done!

**Consistency Experiment:**

NeurIPS has a long history of experimentation. In 2014, NeurIPS ran an experiment in which 10% of submissions were reviewed by two independent committees to quantify the randomness in the review process. This year, we repeated a variant of this experiment to see how the quality of the review process has changed over time.  This paper was part of the experiment and was therefore assigned to two committees (consisting of reviewers, an Area Chair, and a Senior Area Chair) that reached independent decisions.  If both committees made the same recommendation, this recommendation was followed. If a single committee recommended acceptance, the paper was accepted (with the exception of a few cases in which the other committee identified what we considered a fatal flaw, e.g., an error in a key result).

This copy’s committee reached the following decision: **Accept (Poster)**

The other committee assigned to the paper recommended **Reject**.  You can find the other set of reviews, along with any follow up discussion with the authors here:
https://openreview.net/forum?id=guAXBsPR4tY